# Somatic *SF3B1* hotspot mutation in prolactinomas

Chuzhong Li [1,2,3,4,13], Weiyan Xie [1,13], Jared S. Rosenblum [5], Jianyu Zhou[6], Jing Guo[1,2], Yazhou Miao[1,2], Yutao Shen[1,2], Hongyun Wang[1], Lei Gong[1], Mingxuan Li[1,2], Sida Zhao[1,2], Sen Cheng[1,2], Haibo Zhu[1,2], Tao Jiang [6,7], Shiying Ling[8], Fei Wang[8], Hongwei Zhang[9], Mingshan Zhang[9], Yanming Qu[9], Qi Zhang[5], Guilin Li[10], Junmei Wang[10], Jun Ma[11], Zhengping Zhuang [5,12 ✉] & Yazhuo Zhang [1,2,3,4 ✉]

The genetic basis and corresponding clinical relevance of prolactinomas remain poorly understood. Here, we perform whole genome sequencing (WGS) on 21 patients with prolactinomas to detect somatic mutations and then validate the mutations with digital polymerase chain reaction (PCR) analysis of tissue samples from 227 prolactinomas. We identify the same hotspot somatic mutation in splicing factor 3 subunit B1 (*SF3B1^{R625H}*) in 19.8% of prolactinomas. These patients with mutant prolactinomas display higher prolactin (PRL) levels ($p = 0.02$) and shorter progression-free survival (PFS) ($p = 0.02$) compared to patients without the mutation. Moreover, we identify that the *SF3B1^{R625H}* mutation causes aberrant splicing of estrogen related receptor gamma (ESRRG), which results in stronger binding of pituitary-specific positive transcription factor 1 (Pit-1), leading to excessive PRL secretion. Thus our study validates an important mutation and elucidates a potential mechanism underlying the pathogenesis of prolactinomas that may lead to the development of targeted therapeutics.

[1] Department of Cell Biology, Beijing Neurosurgical Institute, Capital Medical University, Beijing 100070, China. [2] Department of Neurosurgery, Beijing Tiantan Hospital affiliated to Capital Medical University, Beijing 100070, China. [3] China National Clinical Research Center for Neurological Diseases, Beijing 100070, China. [4] Brain Tumor Center, Beijing Institute for Brain Disorders, Beijing 100070, China. [5] Neuro-Oncology Branch, National Cancer Institute, National Institutes of Health, Bethesda, MD 20892, USA. [6] Bioinformatics Division, Department of Computer Science and Technology, BNRIST, Tsinghua University, Beijing 100084, China. [7] Department of Computer Science and Engineering, University of California, Riverside, CA 92521, USA. [8] Department of Neurosurgery, The First Affiliated Hospital of University of Science and Technology of China, Hefei 230001, China. [9] Department of Neurosurgery, Sanbo Brain Hospital, Capital Medical University, Beijing 100093, China. [10] Department of Neuropathology, Beijing Neurosurgical Institute, Capital Medical University, Beijing 100070, China. [11] Department of Neuroimaging, Beijing Tiantan Hospital affiliated to Capital Medical University, Beijing 100070, China. [12] Surgical Neurology Branch, National Institute of Neurological Disorders and Stroke, National Institutes of Health, Bethesda, MD 20892, USA. [13] These authors contributed equally: Chuzhong Li, Weiyan Xie. ✉email: zhengping.zhuang@nih.gov; zhangyazhuo@ccmu.edu.cn

The prevalence of pituitary adenomas (PAs) is ~0.1% in adults and nearly half of them are prolactinomas[1], which are typically characterized by increased PRL secretion with related endocrinological symptoms. Mass effects including headaches and visual field defects may occur with parasellar extension of macroadenomas. A subset of prolactinomas may become aggressive and resist therapies[1], but the mechanism of aggressive biological behavior has not been fully determined.

The genetic causes of PAs remain unclear but studies have reported that they have the low mutational burden, reflected by their benign nature[2]. Only several driver mutations have been identified to date in PAs, including guanine nucleotide-binding protein subunit alpha S (GNAS) in somatotroph adenomas[3] and ubiquitin-specific protease 8 (USP8), USP48 and B-Raf proto-oncogene (BRAF) in corticotroph adenomas[4,5]. In prolactinomas, no high-frequency genetic alterations have been reported, which greatly hindered the understanding of tumor pathogenesis and the development of therapeutic strategies.

Herein, we conduct a genomic analysis of resected prolactinomas using whole-genome sequencing (WGS) and identify a hotspot mutation in SF3B1, a component of the U2 small nuclear ribonucleoproteins (snRNP) complex, which has been implicated in other cancer types[6–8]. We then validate the mutation in an additional set of prolactinomas. We demonstrate herein that the identified mutation of SF3B1 results in aberrant splicing of ESRRG—a member of the estrogen receptor-related receptor (ESRR) family—leading to abnormal PRL secretion and tumorigenesis in prolactinomas. This finding may represent a distinct genotype of prolactinomas with unique clinical implications and contributes to the understanding of the molecular mechanism of prolactinomas.

## Results

**Identification of hotspot $SF3B1^{R625H}$ mutation in prolactinomas.** We performed WGS on an initial set of 21 prolactinomas to detect somatic mutations. Ninety somatic mutations in 88 genes were identified using WGS (median, 4; range, 0–21) in the initial patient set ($n = 21$) (Fig. 1a; Supplementary Table 1, 2). Only SF3B1 was found to be mutated in more than one sample ($n = 2$; c.G1874A; p.R625H), which was confirmed by Sanger sequencing (Supplementary Fig. 1). Examination of the SF3B1 coding sequence showed no additional mutations in the gene. The mutation in SF3B1 was validated with digital PCR analysis of 227 prolactinomas including the 21 cases.

Microfluidic-chamber-based digital PCR analysis performed on $SF3B1^{R625H}$ in the prolactinoma tumor samples obtained from an additional 178 prolactinomas (validation set 1) found 38 $SF3B1^{R625H}$ mutations (Fig. 1b). Of the 38 samples, 11 available paired blood samples showed no mutation. Including the initial patient set ($n = 21$), 20.1% (40/199) had the mutation. The results in the main study group were further validated in an independent medical center group of prolactinomas (validation set 2, $n = 28$) (Fig. 1c), in which $SF3B1^{R625H}$ was found in five samples (17.9%) by PCR analysis of c.G1874A (p.R625H) (Supplementary Table 3). Thus, the total number of $SF3B1^{R625H}$ mutant prolactinomas identified was 45/227, 19.8%. Overview on the patients included in this study and summary of SF3B1 mutational status of these patients are shown in Supplementary Table 1.

To study the prevalence of $SF3B1^{R625H}$ mutation in other types of PAs, we screened an additional 154 PAs. There were no $SF3B1^{R625H}$ mutation in such pathological types of PAs ($n = 120$), including 18 thyrotroph adenomas, 33 somatotroph, 30 gonadotroph, 24 corticotroph, and 15 null cell adenomas. In the remaining 34 tumors, 1 of 16 plurihormonal (6.3%) and 1 of 18 mammosomatotroph/mixed adenomas (5.6%) showed the

mutation. These two tumors with the mutation both have a positive immunostaining for PRL. Thus, the $SF3B1^{R625H}$ mutation was only detected in PRL immune-positive PAs.

SF3B1 is the most frequently mutated spliceosome gene, which has an oncogenic role by alternative splicing of pre-mRNAs, resulting in complexity and flexibility in gene expression[9,10]. Phylogenetic analysis of $SF3B1^{R625H}$ locus across species indicated a high conservation level (Supplementary Fig. 2), suggesting an essential role in gene expression and post-transcriptional regulation. Mutations of SF3B1 in other diseases are typically located in HEAT (Huntingtin, elongation factor 3, protein phosphatase 2 A and the yeast PI3-kinase TOR1) repeat domains[6,8,11]. The R625 residue found herein mutated is located in the fourth HEAT repeat (Supplementary Fig. 2). R625 mutations in SF3B1 have been described in uveal melanoma, vulvovaginal mucosal melanoma, and other cancers[8,12,13]. The location of the mutation we identified in these prolactinomas, consistent with mutations in other cancers, suggests the mutation significantly alters function based on its location.

**Impact of $SF3B1^{R625H}$ on prolactinoma function.** To assess the role of $SF3B1^{R625H}$ in prolactinomas, we established primary cell cultures of surgically resected human prolactinomas. We assessed PRL secretions in culture media from mutant and wild-type culture tumor cells. The PRL level in the cell supernatant of the mutant group was higher than the wild type (Fig. 2a). Involvement of endogenous SF3B1 was tested by using short interfering RNA (siRNA); knockdown efficiency was verified by quantitative Real-time PCR (qRT-PCR) and western blot (Fig. 2b, Supplementary Fig. 3). SF3B1 siRNA resulted in decreased PRL secretion in primary culture prolactinoma cells (Fig. 2c). Further, we infected the primary culture prolactinoma cells with adenovirus with $SF3B1^{WT}$ and $SF3B1^{R625H}$. SF3B1 mutation resulted in a significant augment of PRL secretion (Fig. 2d). These results demonstrated increasing PRL secretion with this mutation in human prolactinoma cells.

We then investigated the role and function of $SF3B1^{R625H}$ mutation in the development of prolactinomas by performing colony formation, cell counting kit-8 (CCK-8) and flow cytometry on rat GH3/MMQ pituitary cells with ectopic gene expression at the indicated multiplicity of infection (MOI) (Supplementary Fig. 4) and stable cell line, respectively. Over-expression of $SF3B1^{R625H}$ by adenovirus improved the focus formation of GH3 and MMQ cells (Fig. 2e, h and Supplementary Fig. 5a). After infection of adenovirus expressing $SF3B1^{WT}$ and $SF3B1^{R625H}$, respectively, CCK-8 assay was executed to detect the effect of $SF3B1^{R625H}$ on the proliferation. These results showed $SF3B1^{R625H}$ increased the growth of GH3 and MMQ cells compared with control (Fig. 2f, i). To determine the influence of $SF3B1^{R625H}$ on the apoptosis of GH3 and MMQ cells, we conducted Annexin V/propidium iodide (PI) staining and flow cytometry. The results revealed that the percentages of early and late apoptosis of GH3 and MMQ cells were decreased by $SF3B1^{R625H}$ compared with control (Fig. 2g, j and Supplementary Fig. 5c). Then, we confirmed these findings by a stable GH3 cell line established with the infection of a lentivirus containing $SF3B1^{WT}$ and $SF3B1^{R625H}$ (Fig. 2k, l, m, and Supplementary Fig. 5b, c). Considered together, these data indicate that $SF3B1^{R625H}$ enhances proliferation and suppress apoptosis of GH3 and MMQ cells.

**$SF3B1^{R625H}$ induces aberrant splicing of ESRRG.** RNA-seq and rMATS analysis found 112 significantly different splicing events (98 genes) between prolactinomas with or without SF3B1 mutation (Supplementary Table 4). This included 35 alternative 3′

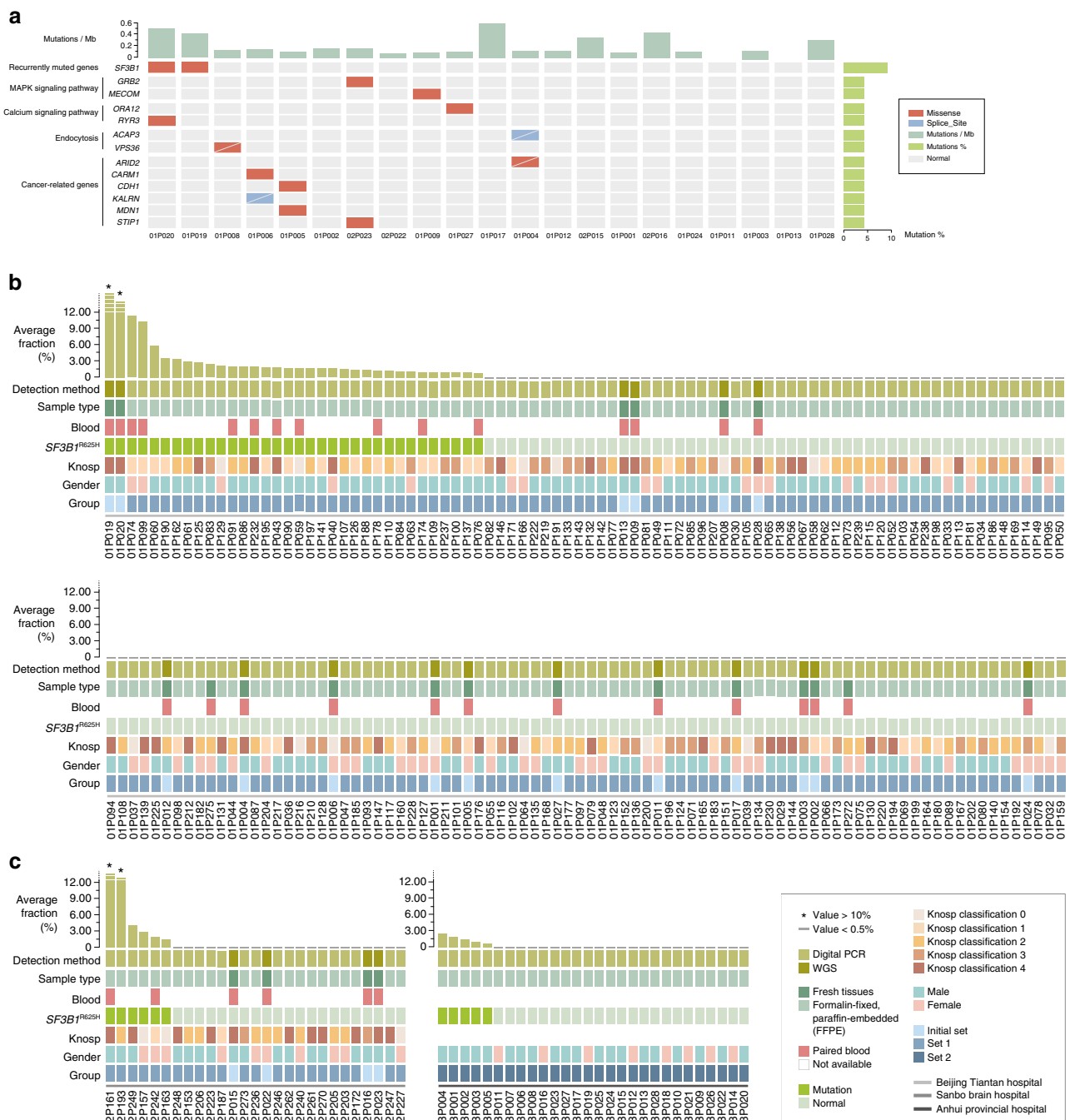

**Fig. 1 Recurrent *SF3B1* Mutations in prolactinomas. a** The mutational landscape of 21 prolactinomas. Samples are displayed in columns from left to right. Each row represents a gene. The rates of synonymous and non-synonymous mutations are expressed in the number of mutations per megabase (Mb) and are displayed in the top panel. The somatic mutation frequencies for each candidate gene are plotted on the right panel. Mutations that were not validated through Sanger sequencing and time of flight mass spectrometer (TOF), because of an unsuccessful amplification or lack of remaining tissues, are represented by a white slash. Mutation types are color-coded as indicated above the image. All candidate genes are considered capable of expression with FPKM of over 1 in >20% of all RNA samples. **b, c** Detection of *SF3B1^R625H* mutations in the prolactinoma tumor samples. The top chart shows the fractional abundance of variants on microfluidic-chamber-based digital PCR analysis in prolactinoma tissue samples. The samples are shown according to the order of highest to lowest frequency, **b** the section 1 included 172 patients from Beijing Tiantan Hospital affiliated to Capital Medical University, **c** the section 2 included 27 patients from Sanbo Brain Hospital and section 3 included 28 patients from the First Affiliated Hospital of University of Science and Technology of China. At the bottom of the chart shows the details of the sample, including the detection method, the sample type, the presence or absence of a paired blood sample, *SF3B1^R625H* mutation detected, the Knosp classification, gender and group. Source data are provided as a Source Data file.

splice sites (A3SS), 2 alternative 5′ splice sites (A5SS), 22 mutually exclusive exons, 36 skipped exons, and 17 retained introns. A significant overrepresentation of regulated A3SS was found in the comparison of *SF3B1^R625H* with wild-type tumors (Supplementary Fig. 6a).

Fig. 6a). These genes all showed upstream cryptic 3′ splice sites with the cryptic AG site located between 16 and 25 nucleotides upstream of the canonical site (Supplementary Fig. 6b). These data reflect the importance of structural similarity in recognizing the

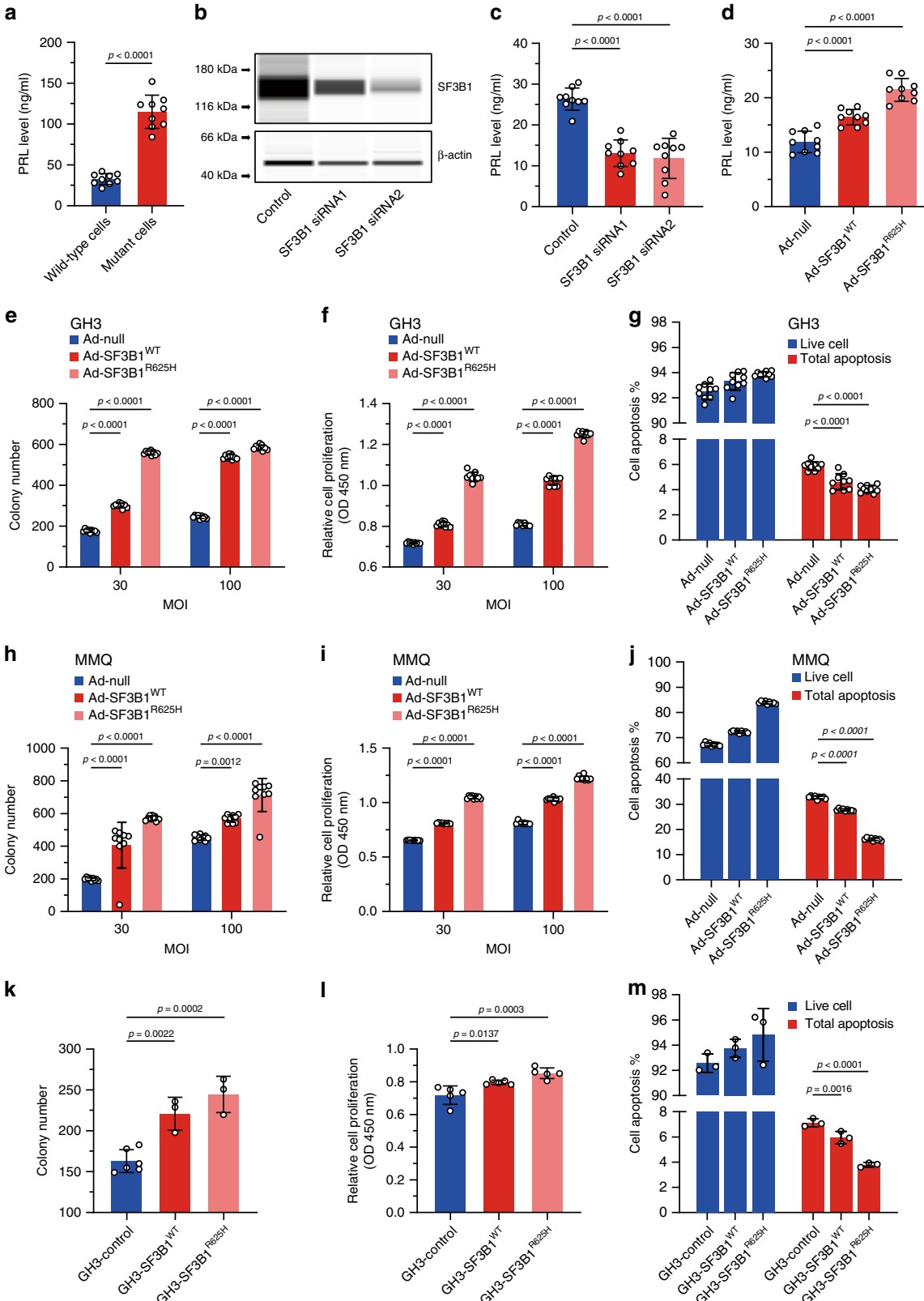

splicing sites in SF3B1, in accordance with the known function of SF3B1 in the recognition of branch-points and 3′ splice sites[12]. Moreover, samples with *SF3B1R625H* showed a set of unique aberrantly spliced junctions (Fig. 3a, Supplementary Fig. 6c). We selected top 20 significant aberrant events (Supplementary Table 5) and the involved gene targets in mutant and wild-type

tumors using the rMATS pipeline and further validated them using RT-PCR, 15 events (14 genes) of which were verified (Supplementary Fig. 7).

SF3B1 is an essential component of the U2 snRNP complex, which binds pre-mRNA and carries sequence-specific RNA-binding activity via the U2 auxiliary factor 2 (U2AF2) association

**Fig. 2 Downstream effects of the *SF3B1* mutation. a** PRL secretion in *SF3B1* wild type and *SF3B1* mutant primary human prolactinomas cells. **b** Representative western blot for SF3B1 expression levels in primary human prolactinoma cells transfected with control or specific SF3B1 siRNA. β-actin was used as internal control. **c** Suppression of the PRL secretion in primary human prolactinoma cells after the SF3B1 knockdown using SF3B1 siRNA. **d** PRL secretion in primary human prolactinoma cells infected with Ad-null, Ad-SF3B1^WT and Ad-SF3B1^R625H are shown, respectively. **e** Focus formation was assessed in GH3 cells infected with Ad-null, Ad-SF3B1^WT, and Ad-SF3B1^R625H at the indicated MOI. **f** Results of CCK-8 cell proliferation assay showing in GH3 cells infected with Ad-null, Ad-SF3B1^WT, and Ad-SF3B1^R625H at the indicated MOI. **g** Annexin V/PI staining and flow cytometry showed the percentages of apoptosis of the GH3 cells infected with Ad-null, Ad-SF3B1^WT, and Ad-SF3B1^R625H. **h** Focus formations were assessed in MMQ cells infected with Ad-null, Ad-SF3B1^WT, and Ad-SF3B1^R625H at the indicated MOI. **i** Results of CCK-8 cell proliferation assay showing in MMQ cells infected with Ad-null, Ad-SF3B1^WT, and Ad-SF3B1^R625H at the indicated MOI. **j** Annexin V/PI staining and flow cytometry showed the percentages of apoptosis of the MMQ cells infected with Ad-null, Ad-SF3B1^WT, and Ad-SF3B1^R625H. **c–i** $n = 9$ per group. **k** Focus formations were assessed in stable GH3-control, GH3-SF3B1^WT, GH3-SF3B1^R625H cells ($n = 6$ for GH3-control and $n = 3$ for two other cells). **l** Results of CCK-8 cell proliferation assay showing in stable GH3-control, GH3-SF3B1^WT, GH3-SF3B1^R625H cells ($n = 5$ per group). **m** Annexin V/PI staining and flow cytometry showed the percentages of apoptosis of stable GH3-control, GH3-SF3B1^WT, GH3-SF3B1^R625H cells. Data are represented as mean ± SD. The *p* value by two-tailed unpaired *t* test is indicated in **a**. The *p* values by one-way ANOVA followed by Dunnett's multiple comparisons test in **c**, **d**, **k**, **l**, **m** and followed by Tukey's multiple comparisons post hoc test in **e**, **f**, **g**, **h**, **i**, **j** are indicated. Source data are provided as a Source Data file.

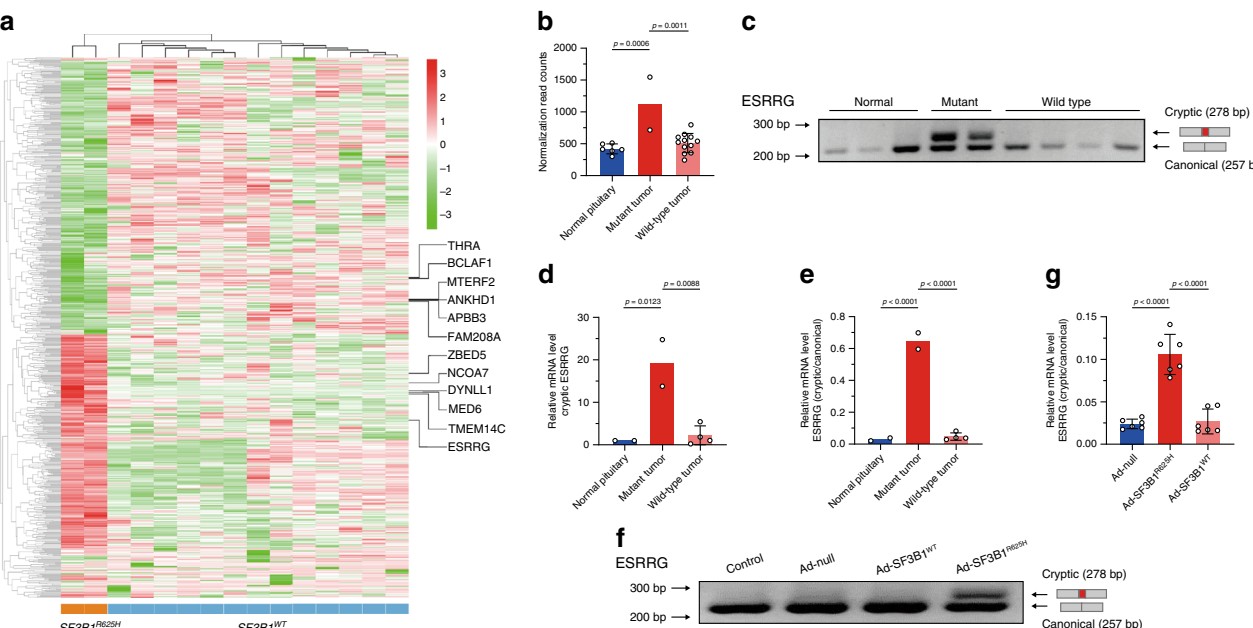

**Fig. 3 Aberrant splicing of ESRRG in *SF3B1* mutant prolactinoma samples. a** Heatmap of *SF3B1*^R625H-induced splicing alterations in prolactinomas. Rows represent alternative spliced events while columns represent patients. The heatmap matrix values indicate percent spliced in (PSI) value for each individual splicing event to quantify the level of inclusion isoform, which is versus the total abundance of all isoforms, normalized as *Z* scores. Genes listed on the right side are the A3SS splicing genes verified by RT-PCR. **b** The graphs show the ESRRG expression from RNA-seq data in normal pituitaries, prolactinoma samples with mutant and wild type *SF3B1* (normal pituitary $n = 6$, mutant tumor $n = 2$, and wild-type tumor $n = 12$). **c** RT-PCR validation of aberrant splicing of ESRRG in *SF3B1* mutant prolactinoma samples. The higher 278 bp band corresponds to the cryptic ESRRG transcript and the lower 257 bp band corresponds to the canonical ESRRG transcript. **d** The graphs show expression levels of the cryptic ESRRG in normal pituitaries, *SF3B1* mutant and *SF3B1* wild-type prolactinoma samples (normal pituitary $n = 2$, mutant tumor $n = 2$, and wild-type tumor $n = 4$). **e** The qRT-PCR graphs show cryptic to canonical isoform ratios in the *SF3B1* mutant prolactinoma samples (normal pituitary $n = 2$, mutant tumor $n = 2$, and wild-type tumor $n = 4$). **f** RT-PCR validation of aberrant splicing of ESRRG in primary human prolactinoma cells infected with Ad-null, Ad-SF3B1^R625H, and Ad-SF3B1^WT respectively. The higher 278 bp band corresponds to the aberrant ESRRG transcript and the lower 257 bp band corresponds to the canonical ESRRG transcript. **g** The qRT-PCR graphs show cryptic to canonical isoform ratio in primary human prolactinoma cells infected with Ad-null, Ad-SF3B1^R625H, and Ad-SF3B1^WT, respectively ($n = 6$ per group). Results are expressed as mean ± SD. The *p* values by one-way ANOVA followed by Dunnett's multiple comparisons test in **b** and followed by Tukey's multiple comparisons post hoc test in **d**, **e**, **g** are indicated. Source data are provided as a Source Data file.

with the pre-mRNA on the polypyrimidine track[14,15]. RBPmap[16] scan of the U2AF2-binding motif (from CISBP-RNA[17], id: M077_0.6) on the 15 proven alternatively spliced events demonstrated that all 14 genes had the binding motifs recognized. ESRRG contained the most binding motifs (3088, significantly more than others), and over half of the annotated introns ended with the U2AF2-binding motif (Supplementary Fig. 8a, b). Interrogation of the location distribution of U2AF2-binding motif on ESRRG found that the U2 snRNP was

potentially related to the abnormal splicing of ESRRG (Supplementary Fig. 8c). This suggested the impact of SF3B1^R625H on the splicing of ESRRG through association with the U2 snRNP. ESRRG belongs to the ESRR family, which is closely associated with the ER family and has common target genes, co-regulatory factors, and promoters[18], whereas ER is an established regulatory factor of PRL synthesis and lactotroph proliferation[19–21] and binds to a single estrogen response element located within the distal PRL enhancer.

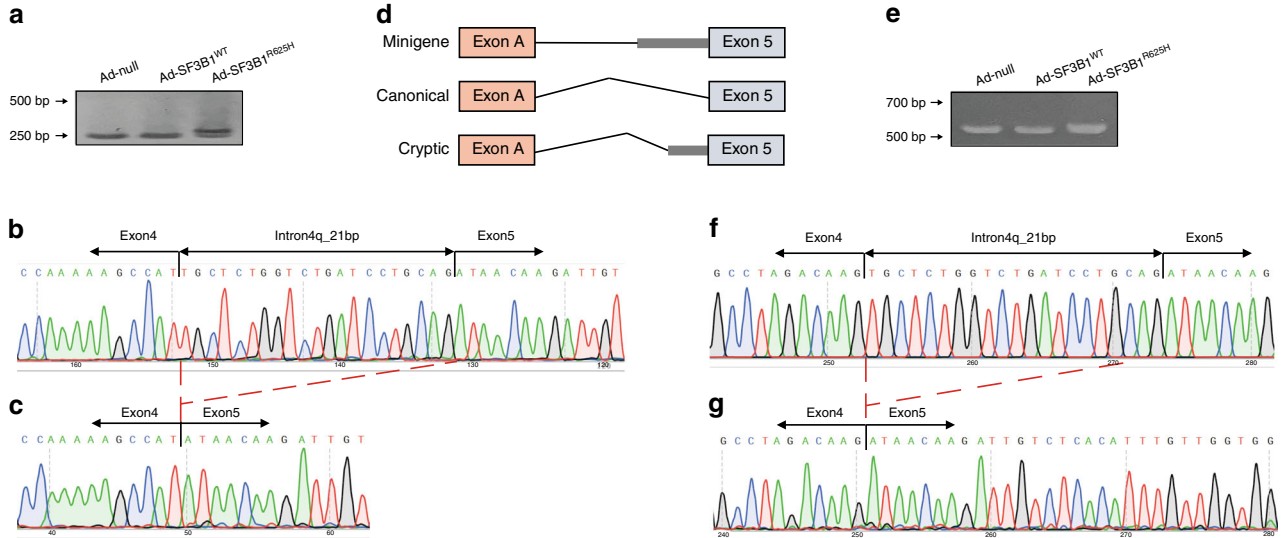

**Fig. 4 Alternative splicing assays in minigene splicing reporters. a** RT-PCR validation of aberrant splicing of ESRRG in MCF7 cells transduced with Ad-null, Ad-SF3B1$^{WT}$, and Ad-SF3B1$^{R625H}$, respectively. The higher band corresponds to the cryptic ESRRG transcript and the lower band corresponds to the canonical ESRRG transcript. **b**, **c** Sanger sequence of gel-purified fragment showing the aberrant ESRRG transcript in SF3B1$^{R625H}$ cells **b**, the canonical ESRRG transcript in SF3B1$^{WT}$ cells **c**. The red dotted lines indicate the location of 21 nucleotides. **d** Structure of minigene splicing reporters spanning the E5 region of ESRRG. **e** RT-PCR validation of aberrant splicing of ESRRG in MCF7 cells co-transduced minigene reporter with Ad-null, Ad-SF3B1$^{WT}$ and Ad-SF3B1$^{R625H}$, respectively. The higher band corresponds to the cryptic ESRRG transcript and the lower band corresponds to the canonical ESRRG transcript. **f**, **g** Sanger sequence of gel-purified fragment showing the aberrant ESRRG transcript in SF3B1$^{R625H}$ cells **f**, the canonical ESRRG transcript in SF3B1$^{WT}$ cells **g**.

RNA-seq showed significantly increased expression of ESRRG in SF3B1$^{R625H}$ samples (Fig. 3b), which was not observed in wild-type tumors and normal pituitaries. In the transcript-level analysis, we found that the main contribution of the high expression of ESRRG in mutant groups was from transcript NM_001243518.1 (hereafter, referred to as cryptic ESRRG transcript) (Supplementary Fig. 8d, e). It was consistent with the results of rMATS that only the cryptic ESRRG transcript contained the aberrant splicing event detected by rMATS. Alternative splicing of ESRRG was significantly affected in the SF3B1 mutant samples, confirmed by RT-PCR (Fig. 3c). The cryptic ESRRG transcript was only observed in the SF3B1 mutant samples, but not in wild-type or normal pituitaries, observed by qRT-PCR experiments (Fig. 3d, e). To confirm the regulation of SF3B1$^{R625H}$ on ESRRG, we infected the adenovirus carrying SF3B1$^{R625H}$ mutation in primary cultured tumor cells, and the results showed that the cryptic ESRRG transcript was only observed in the Ad-SF3B1$^{R625H}$ group (Fig. 3f). The qRT-PCR results validated that SF3B1$^{R625H}$ could induce a high cryptic to canonical isoform ratio in ESRRG in primary human prolactinoma cells (Fig. 3g). The same results were also observed in human MCF7 cells (human breast adenocarcinoma cell) (Fig. 4a). Sanger sequence analysis showed that this was the expected fragment with a 21 bp elongation of exon 5 (Fig. 4b, c). We then investigated the alternative splicing using a minigene splicing reporter system. The minigene construct contains a fragment of the ESRRG gene spanning exons 5 and includes 353 bp of intron four sequences in this region (Fig. 4d). After infection of adenovirus expressing SF3B1$^{WT}$ and SF3B1$^{R625H}$, respectively, this reporter was spliced to produce two major products when co-transfected into MCF7 cells: a fully spliced RNA containing exons 5 (Fig. 4e, left and middle lanes), and a larger transcript that retained extra 21 bp (Fig. 4e, right lane). Sanger sequencing analysis showed that these were the expected fragment with extra 21 bp (Fig. 4f, g). The minigene results suggest that the R625H mutation of SF3B1 does induce aberrant splicing of ESRRG. The above results indicate that the ESRRG splicing pattern is sensitive to the SF3B1 gene function.

**SF3B1 interacts directly with ESRRG mRNA.** To further understand the mechanism of SF3B1-mediated ESRRG splicing, we examined whether SF3B1 could bind directly to ESRRG. RNA immunoprecipitation (RIP) demonstrated SF3B1 could bind ESRRG mRNA in SF3B1 wild-type prolactinomas (Fig. 5a). In the RIP complex of prolactinomas, we detected the cryptic ESRRG only in SF3B1 mutant prolactinoma samples (Fig. 5b). This result is consistent with the aforementioned cryptic ESRRG only existing in SF3B1 mutant samples. This confirmed that the SF3B1$^{R625H}$ would trigger the aberrant splicing in ESRRG.

Then, we analyzed the endogenous binding of SF3B1 to ESRRG via a cross-linking immunoprecipitation and quantitative PCR (CLIP-qPCR) assay[22] in MCF7 cells, using 15 pairs of primers with overlapping 100 bp amplicons, which allowed detection of the protected ESRRG mRNA segments bound by SF3B1 and the mapping of SF3B1-binding sites on ESRRG at 100-nt intervals (Fig. 5c). Two major peaks were detected, suggesting that ESRRG contains two SF3B1-binding sites, respectively, in the P12 and P14 segments (Fig. 5c). The results further narrowed down the major SF3B1-binding motif in the sequence around P12 and P14 segments of ESRRG.

In order to compare the binding ability of wild-type and mutant SF3B1 with ESRRG and confirm the binding sites of mutant SF3B1 and cryptic ESRRG, we conducted CLIP experiments in the wild-type and SF3B1 mutant prolactinomas tissue samples, respectively. As shown in Fig. 5d, consistent with the results of MCF cells, the P12 and P14 segments of ESRRG showed the highest binding ability to SF3B1 in wild-type prolactinomas. Compared with the canonical ESRRG, the cryptic ESRRG had an extra 21 bp of the exon 5, so we designed primers for this unique 21 bp in cryptic ESRRG (P8, only exists in cryptic ESRRG sequence) and corresponding fragment (P18, only exists in canonical ESRRG sequence) in canonical ESRRG. The results (Fig. 5e) showed that in the SF3B1 mutant prolactinomas samples, P8 segment instead of P12/P14 had the highest enrichment value, which reached 1016, far higher than the second 95 (P15). The enrichment value of P8 in cryptic ESRRG is

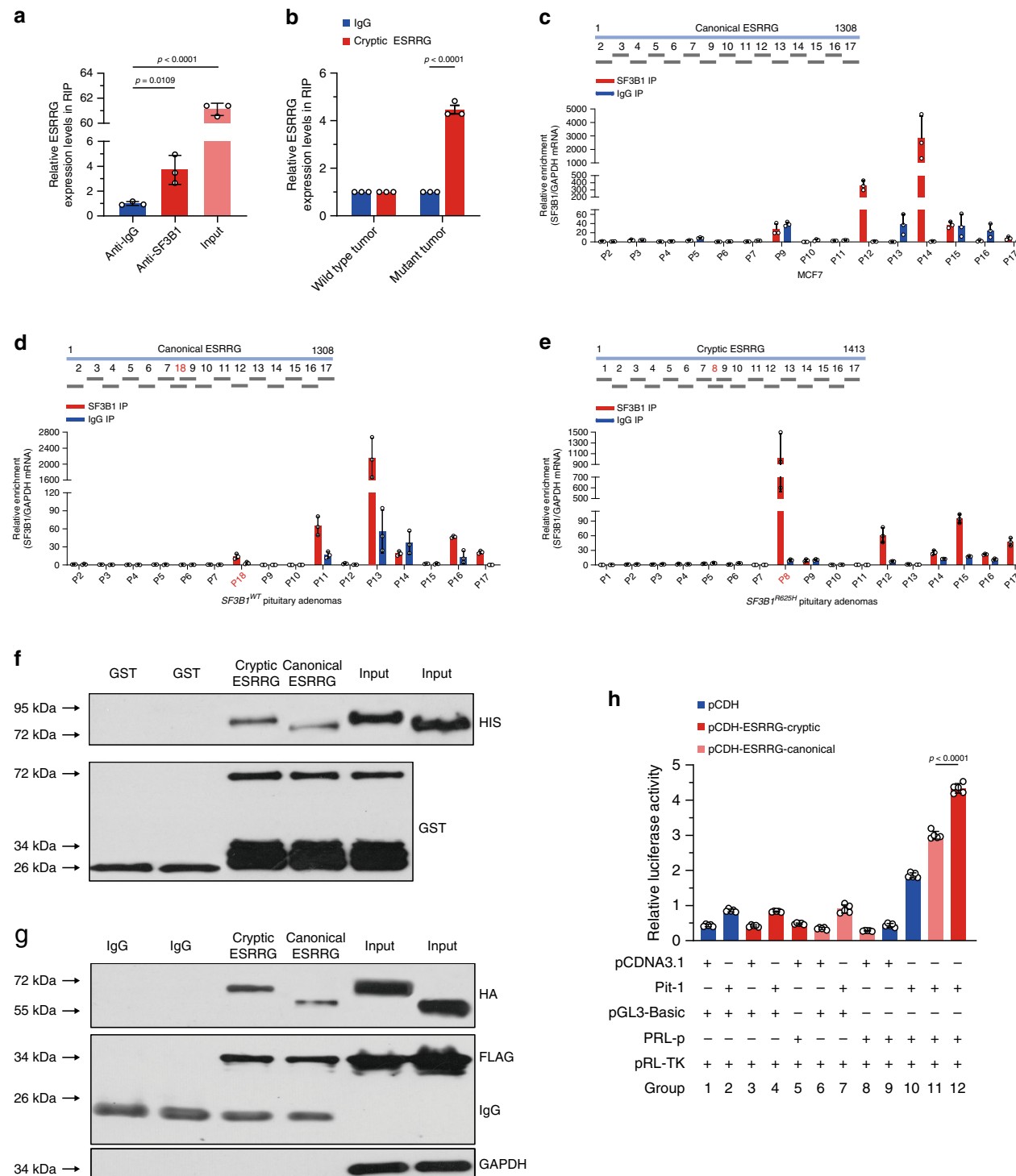

very high, whereas the enrichment of corresponding P18 in canonical ESRRG is not obvious. These results suggest that the mutant SF3B1 has a strong binding ability with the extra 21 bp of cryptic ESRRG.

**Cryptic ESRRG binds Pit-1 with greater affinity.** Pit-1 is crucial for the pituitary-specific expression of the *PRL* gene[23–25]. To determine whether there is a direct interaction between Pit-1 and ESRRG, we immunoprecipitated Pit-1/ESRRG from human PA lysates and probed with ESRRG/Pit-1 antibody, respectively. The

results indicate a physical interaction between Pit-1 and ESRRG in human PA (Supplementary Fig. 9). Then we examined Pit-1 protein binding in relation to ESRRG alternative splicing status. Pit-1 protein fused to glutathione S-transferase (GST) pull-down HIS-tagged cryptic ESRRG, whereas it bound weakly to canonical ESRRG (Fig. 5f). We next co-expressed Flag-tagged Pit-1 HA-tagged ESRRG in human embryonic kidney cells 293 (HEK293) cells and immunoprecipitated them with anti-Flag antibody. As determined by co-immunoprecipitation, cryptic ESRRG binds strongly to Pit-1, whereas canonical ESRRG retained weak binding (Fig. 5g). HEK293 cells were measured by luciferase

**Fig. 5 The effects of aberrant splicing on ESRRG. a** ESRRG mRNA levels in immunoprecipitates were determined through qRT-PCR analysis ($n = 3$ per group). ESRRG mRNA expression levels were presented as fold enrichment ratios compared with IgG. **b** Cryptic ESRRG mRNA levels in immunoprecipitates of wild-type and mutant primary human prolactinoma cells were determined by qRT-PCR ($n = 3$ per group). Expression levels of ESRRG mRNA were presented as fold enrichment ratios compared with IgG. **c–e** CLIP of SF3B1-bound ESRRG mRNA in MCF7 cells **c**, $SF3B1^{WT}$ prolactinomas **d**, or $SF3B1^{R625H}$ prolactinomas **e** ($n = 3$ per group). qPCR was used to identify the region in ESRRG bound by SF3B1 protein. The amount of immunoprecipitated RNAs in each sample is represented as a signal relative to the negative (IgG) sample. Schematic representation of human ESRRG segments amplified by primer pairs for CLIP-qPCR. **f** GST-Pit-1 fusion protein immobilized on glutathione beads and incubated with HIS-tagged cryptic or canonical ESRRG proteins. Bound ESRRG proteins were detected by anti-HIS immunoblotting. **g** Immunoblotting with the indicated antibodies of FLAG immunoprecipitated from lysates of HEK293 cells co-transfected with FLAG-tagged Pit-1, HA-tagged cryptic, or canonical ESRRG. **h** Relative luciferase activity of *PRL* promoter in HEK293 cells transfected with pCDNA3.1-Pit-1 (Pit-1), pGL3-basic-PRL promoter (PRL-p), or pCDH-ESRRG-canonical/cryptic and their corresponding empty vectors including pCDNA3.1, pGL3-basic, and pCDH. pRL-TK was used as a control ($n = 4$ per group). Results are expressed as mean ± SD. The $p$ values by one-way ANOVA followed by Dunnett's multiple comparisons test in **a** and followed by Bonferroni's multiple comparisons test in **h** are indicated. The $p$ value by two-tailed unpaired $t$ test is indicated in **b**. Source data are provided as a Source Data file.

reporter assay with transiently co-expressed Pit-1, *PRL* promoter, cryptic, and canonical ESRRG clones. The cryptic ESRRG demonstrated stronger *PRL* transcriptional activation than the canonical ESRRG (Fig. 5h). Thus, these results demonstrate that cryptic ESRRG has a stronger affinity to bind to the Pit-1, potentially resulting in increased activation of *PRL* transcription.

**The effects of aberrant splicing on ESRRG**. The reduced expression of ESRRG by siRNA (70–80% reduced) in transduced primary cultured prolactinoma cells resulted in a significant reduction of PRL secretion (Fig. 6b) in culture media, confirmed by qRT-PCR and western blot (Fig. 6a, Supplementary Fig. 10). Further, we infected the primary culture prolactinoma cells with adenovirus with canonical and cryptic ESRRG, respectively. Cryptic ESRRG resulted in a significant augment of PRL secretion (Supplementary Fig. 11). These results indicate that the cryptic ESRRG, resulting from $SF3B1^{R625H}$ contributes to an excess of PRL activation in prolactinomas, beyond physiologic with the canonical ESRRG in the pituitary.

We then investigated the effect of ESRRG on the prolactinoma cell proliferation and growth. After infection of adenovirus expressing canonical ESRRG and cryptic ESRRG, respectively, CCK-8 assay was performed to detect the effect of ESRRG on the proliferation. These results showed that ESRRG increased the growth of GH3 and MMQ cells compared with control (Fig. 6c, d). Both canonical ESRRG and cryptic ESRRG overexpression by adenovirus can improve the focus formation of GH3 and MMQ cells (Fig. 6e, f and Supplementary Fig. 12). PI staining and flow cytometry results revealed that the percentages of early and late apoptosis of GH3 and MMQ cells were decreased by canonical ESRRG and cryptic ESRRG compared with control (Fig. 6g, h, Supplementary Fig. 13). Considered together, these data indicate that ESRRG, especially cryptic ESRRG enhances proliferation and suppresses apoptosis of GH3 and MMQ cells.

**Clinical relevance of SF3B1 mutation**. Clinical characteristics of the 227 patients with prolactinomas were analyzed, 45 of whom contained $SF3B1^{R625H}$ mutation. Clinical features including age, tumor size, and tumor invasion showed no significant difference between the two groups. However, gender analysis revealed that there was a significant male preference in the mutant population ($p = 0.02$, Pearson's $\chi^2$ test; Fig. 7a, Table 1). The frequency of $SF3B1^{R625H}$ mutation in male patients was 24.34% compared with 10.67% in female patients. Patients with mutant SF3B1 showed higher levels of PRL (plasma PRL/tumor size), as compared with the wild-type group. This suggested that the SF3B1 mutant prolactinomas have higher PRL production than the wild-type group ($p = 0.02$, Mann–Whitney $U$ test, Fig. 7b, Table 1). Most importantly, according to the maximum duration of 10-year follow-up data, we observed that the mutated SF3B1 group was

significantly associated with poor progression-free survival (Fig. 7c). In two mutant cases of our discovery cohort, the prolactinomas displayed unusual malignancy. These two patients even suffered from additional surgeries within 2 years because of rapid residual tumor regrowth. Although limited by case number, this evidence supports that $SF3B1^{R625H}$ may contribute to an enhanced malignancy of prolactinomas.

## Discussion

The driving genetic events that contribute to prolactinoma tumorigenesis remain unknown. We performed genomic analysis of 21 prolactinomas by WGS and recurrent $SF3B1^{R625H}$ mutation was found in two tumors. We performed digital PCR in a validation set of 227 prolactinomas and identified 45 tumors (19.8%) with the hotspot $SF3B1^{R625H}$ mutation. The mutation was not detected in other types of PAs besides two mixed PAs with positive PRL immunohistochemical staining. Therefore, the $SF3B1^{R625H}$ mutation appears to be a unique genetic signature of PRL-secreting adenomas.

SF3B1 has an important oncogenic role in the pathogenesis and development of many tumors. Those reported SF3B1 mutations are nearly all located in the C-terminal HEAT domains (residues 622–781). For instance, E662D, K666Q, K700E, and G724D were common in CLL[11]. In uveal melanomas, although R625 was the predominant mutation, several forms including R625C, R625G, R625H, and R625L could be detected[8].

Here, we demonstrated that our subset of prolactinomas harbored $SF3B1^{R625H}$ mutation, which has been predicted to be more deleterious and probably oncogenic[6]. We then confirmed that this mutation resulted in a gain-of-function of ESRRG through alternative splicing. ESRRG belongs to the ESRR family whose biological functions are involved in estrogen-signaling pathways. We found that aberrant ESRRG is transcriptionally active and induces abnormal *PRL* transcription, which suggests that the high levels of PRL in SF3B1 mutant prolactinomas are likely to be caused by this pathway. Data evaluation captured a gender preference in the prolactinoma mutant subset with 24.34% of the total population of males versus 10.67% of females. This is different from the historically observed female gender dominance of prolactinomas[26,27]. The ESRRG mediated mechanism of PRL secretion showed ER independence, supporting the observed gender distribution.

In the present study, we established the role of the recurrent $SF3B1^{R625H}$ mutation in this subset of prolactinomas. The $SF3B1^{R625H}$ mutation promotes PRL hypersecretion through aberrant ESRRG splicing via a stronger affinity for Pit-1, resulting in greater transcriptional activation of PRL (Fig. 7d). The mutation leads to enhanced cell proliferation and decreased apoptosis of prolactinoma cells. The study demonstrated that SF3B1 mutation was significantly associated with poor prognosis in

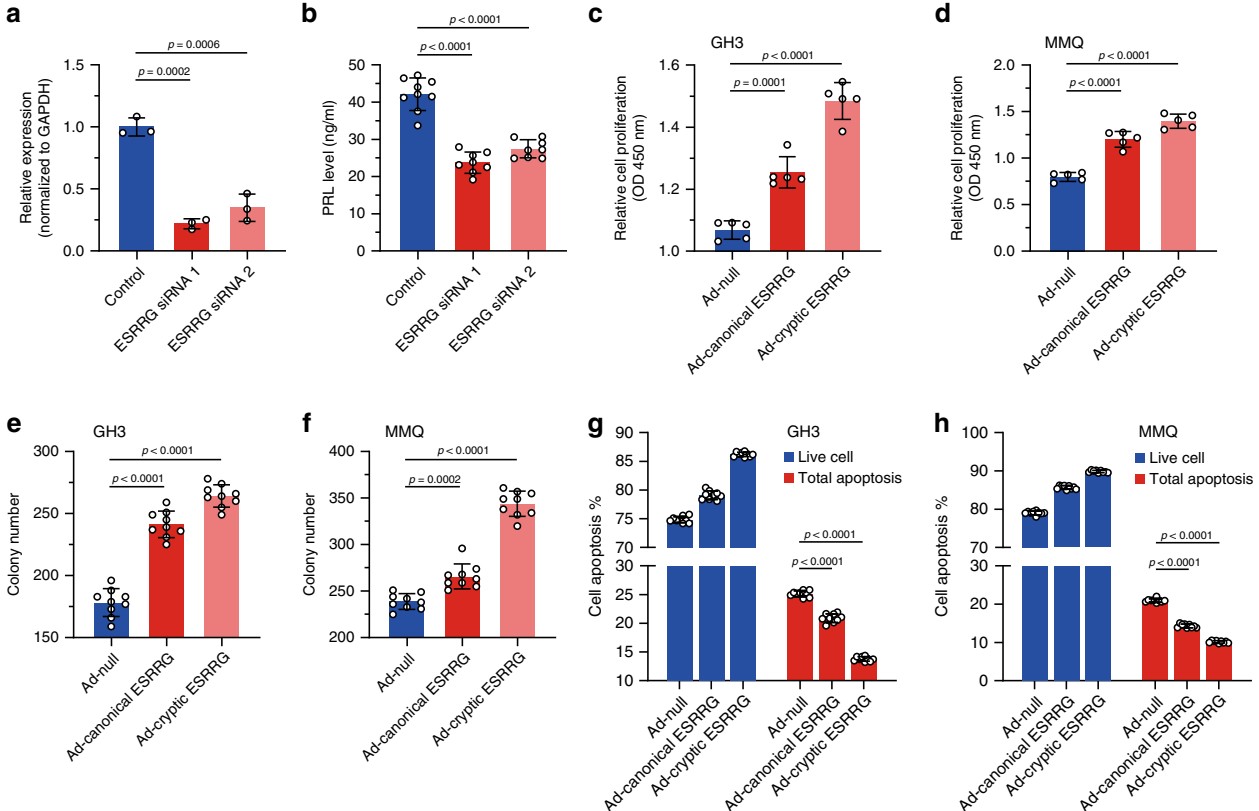

**Fig. 6 Downstream effects on aberrant splicing on ESRRG. a** The qRT-PCR results display ESRRG mRNA expression levels in primary human prolactinoma cells transfected with control or specific ESRRG siRNA. GAPDH was used as an internal control ($n = 3$ per group). **b** Suppression of PRL secretion in primary human prolactinomas cells after ESRRG knockdown using ESRRG siRNA ($n = 9$ per group). **c** Results of CCK-8 cell proliferation assay in GH3 cells infected with Ad-null, Ad-canonical ESRRG, and Ad-cryptic ESRRG ($n = 5$ per group). **d** Results of CCK-8 cell proliferation assay in MMQ cells infected with Ad-null, Ad-canonical ESRRG, and Ad-cryptic ESRRG ($n = 5$ per group). **e** Focus formation was assessed in GH3 cells infected with Ad-null, Ad-canonical ESRRG and Ad-cryptic ESRRG ($n = 9$ per group). **f** Focus formations were assessed in MMQ cells infected with Ad-null, Ad-canonical ESRRG, and Ad-cryptic ESRRG ($n = 9$ per group). **g** Annexin V/PI staining and flow cytometry showed the percentages of apoptosis of the GH3 cells infected with Ad-null, Ad-canonical ESRRG, and Ad-cryptic ESRRG ($n = 9$ per group). **h** Annexin V/PI staining and flow cytometry showed the percentages of apoptosis of the MMQ cells infected with Ad-null, Ad-canonical ESRRG, and Ad-cryptic ESRRG ($n = 9$ per group). Results are expressed as mean ± SD. The p values by one-way ANOVA followed by Dunnett's multiple comparisons test in **a**, **b** and followed by Tukey's multiple comparisons post hoc test in **c**–**h** are indicated. Source data are provided as a Source Data file.

patients with prolactinomas. Although continued postoperative treatment of prolactinomas with dopamine agonists for control of PRL levels owing to possible residual or recurrent tumors is a standard measure, which may make PFS difficult to assess, we believe that this mutation has clinical relevance in defining prognostic subgroups and implications for developing precision therapeutic targeting, as evidenced by our significant correlation herein.

In conclusion, we identified $SF3B1^{R625H}$ as a disease-causative mutation in a subset of prolactinomas and elucidated a cellular mechanism in which alternative splicing of ESRRG pre-mRNAs is constitutively activated and results in estrogen-independent increased PRL section in these tumors. In subsequent studies, we might control excessive PRL secretion in prolactinomas by targeting different alternative splicing form of ESRRG.

## Methods

**Study patients**. We retrospectively reviewed 381 PAs including prolactinomas $n = 227$, other PAs $n = 154$ (18 thyrotroph, 33 somatotroph, 30 gonadotroph, 24 corticotroph, 15 null cell, 16 plurihormonal, and 18 mammosomatotroph/mixed somatotroph and prolactinomas) from patients who underwent surgery for PA at Beijing Tiantan Hospital, Sanbo Brain Hospital Capital Medical University and The First Affiliated Hospital of University of Science and Technology of China. All 227 prolactinoma patients had plasma PRL levels of > 90 ng ml$^{-1}$ and tumor with positive immunostaining for PRL (Supplementary Table 6). All diagnoses of

prolactinomas were confirmed by a multidisciplinary group consisting of neuro-surgeons, neuroradiologists, and neuropathologists (Supplementary Table 7) and normal human anterior pituitaries were obtained from a donation program and the donors died of non-endocrine diseases.

**Study design**. The main study group comprised patients from Beijing Tiantan Hospital affiliated to Capital Medical University. Tissue samples of prolactinomas and peripheral blood samples were obtained from these patients, flash-frozen, and stored at Beijing Neurosurgical Institute, Beijing, China. WGS was performed to detect somatic mutations on the initial patient set ($n = 21$). The results of DNA sequencing were confirmed with microfluidic-chamber-based digital PCR analysis or Sanger sequencing. All procedures performed with the use of samples obtained from patients were approved by the Ethics Committee of Beijing Tiantan Hospital. All the patients signed informed consent.

The remaining 360 patients were screened for mutation validation. The validation group comprised PA patients from Beijing Tiantan Hospital affiliated to Capital Medical University, Sanbo Brain Hospital (validation set 1) and The First Affiliated Hospital of University of Science and Technology of China (validation set 2). Microfluidic-chamber-based digital PCR analysis was performed to detect $SF3B1$ mutations in all sets of patients. DNA samples were obtained from formalin-fixed, paraffin-embedded tissues of PAs from these patients. The PA patient groups can be seen in Supplementary Table 1.

**RNA sequencing**. RNA-seq was performed in two $SF3B1$ mutant tumors, 13 $SF3B1$ wild-type tumors and 10 normal pituitary glands. A total amount of 3 μg RNA per sample was used as input material for RNA sample preparations. First, ribosomal RNA was removed by Epicentre Ribo-zero rRNA Removal Kit (RZH1046, Epicentre). Subsequently, sequencing libraries were generated using the

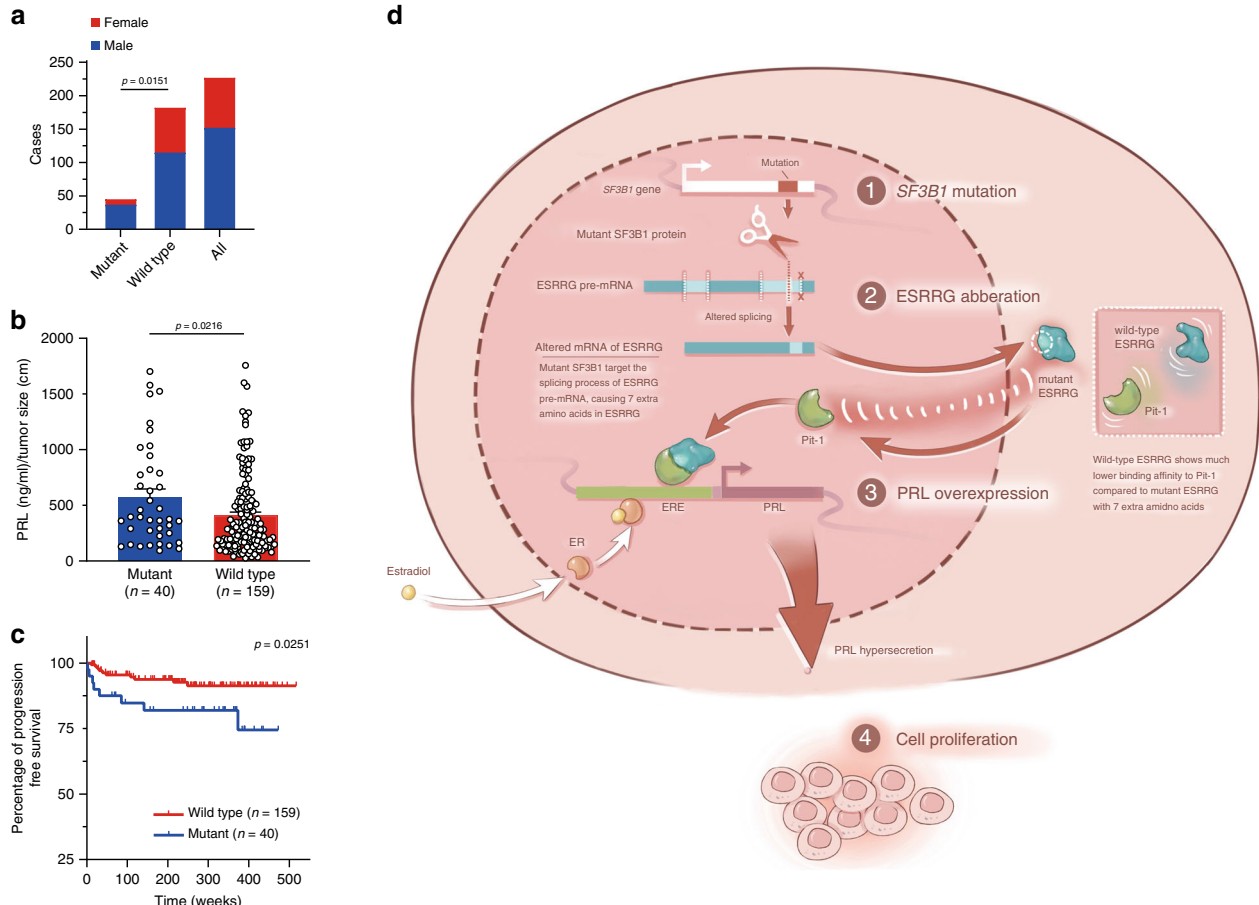

**Fig. 7 Clinical relevance of the *SF3B1* mutation. a** The gender distribution is shown within different groups (mutant group *n* = 45 and wild group *n* = 182). **b** The PRL level (PRL/tumor size) in *SF3B1* mutant and *SF3B1* wild-type prolactinomas are displayed (mutant group *n* = 40 and wild group *n* = 159). Results are given as mean ± SD. **c** Kaplan–Meier survival plots of 199 prolactinomas patients stratified by *SF3B1* mutation status (mutant group *n* = 40 and wild group *n* = 159). **d** Schematic representation showing the proposed mechanisms of how *SF3B1* mutations lead to increased PRL secretion and tumor growth in prolactinomas. The *p* values by chi-square test in **a** and by two-tailed Mann–Whitney test in **b** are indicated. The *p* value by log-rank (Mantel–Cox) test is indicated in **c**. Source data are provided as a Source Data file.

## Table 1 Clinical characteristics of prolactinoma patients.

| Characteristic | Wild type | Mutant | P value |
|---|---|---|---|
| *Age at diagnosis (year)* | | | |
| Median | 38.00 | 39.00 | 0.45[#] |
| Interquartile range | 21.75 | 16.00 | |
| *Sex (case)* | | | |
| Male | 115 | 37 | 0.02* |
| Female | 67 | 8 | |
| *PRL (ng ml⁻¹)/tumor size (cm)‡* | | | |
| Median | 288.12 | 385.58 | 0.02[#] |
| Interquartile range | 398.19 | 552.11 | |
| *Tumor invasion (case)* | | | |
| Non-invasion | 84 | 27 | 0.10* |
| Invasion | 75 | 13 | |
| *Tumor size (cm)‡* | | | |
| Median | 2.80 | 2.60 | 0.74[#] |
| Interquartile range | 1.60 | 2.03 | |

Age and sex data were available for 182 patients with no mutation and 45 patients with the mutation. PRL/tumor size, tumor invasion, and tumor size data were available for 159 patients with no mutation and 40 patients with mutation. In total, 28 cases from the First Affiliated Hospital of University of Science and Technology of China were excluded from statistical analysis because of different PRL dilution measurement protocol.
[#]Values were compared by two-tailed Mann–Whitney *U* test.
*Values were compared by Pearson's $\chi^2$ test.
‡Tumor size: maximal tumor diameter.
Source data are provided as a Source Data file.

rRNA-depleted RNA by NEBNext Ultra Directional RNA Library Prep Kit for Illumina (E7420L, NEB) following the manufacturer's protocol. First strand cDNA was synthesized using random hexamer primer. Second-strand cDNA synthesis and making, which incorporates dUTP into the second strand, converts the cDNA. Double-stranded DNA was repaired via exonuclease/polymerase activities and then added adenylation to the 3′ end. After adapter ligation and library amplification, the library fragments were purified with AMPure XP system (Beckman Coulter, Beverly, USA) in order to select fragments of preferentially 150–200 bp in length. The strand marked with dUTP is not amplified, allowing strand-specific sequencing. At last, products were purified (AMPure XP system) and library quality was assessed on the Agilent Bioanalyzer 2100 system. After cluster generation, the libraries were sequenced on an Illumina Hiseq X platform and 150 bp paired-end reads were generated.

**Alternative splicing analysis.** Alternative 3′ and 5′ splice sites, skipped exons, mutually exclusive exons, and retained introns were quantified using rMATS[28] (http://rnaseq-mats.sourceforge.net/index.html). The default parameters were used for the comparison of two groups, which carrying p. R625H hotspot mutation on *SF3B1* (*SF3B1*^R625H^) and wild-type samples (*SF3B1*^WT^). The alternative spliced events with a false discovery rate < 0.05 and inclusion level difference values > 0.3 or < −0.3 were selected for endpoint reverse transcriptase-PCR (RT-PCR) validation.

The ref. [29] analyzed the cryptic splicing event by setting a cutoff value for the results of rMATS. So, we use a similar criterion to define the cryptic transcripts: we collected data sets from the rMATS output on the basis of FDR < 0.05. Then we selected top 20 significant aberrant events (IncLevelDifference > 0.3 or IncLevelDifference < −0.3) and the involved gene targets in mutant and wild-type tumors using the rMATS pipeline and further validated them using RT-PCR. We define it as cryptic transcripts that dominant transcriptional isoform in mutants, similarly as canonical transcripts in wild type.

**Unsupervised hierarchical clustering of the samples**. Unsupervised clustering of $SF3B1^{R625H}$ and $SF3B1^{WT}$ tumors use the alternative spliced events with $p$ value < 0.05. Taken the batch effect into consideration, we proceeded with unsupervised hierarchical clustering selecting the top 5000 most viable A3SS alternative splicing events to separate and cluster samples.

**Binding motif analysis of U2AF2**. As mentioned in the main content, SF3B1 is an essential component of the U2 snRNP complex, which binds pre-mRNA via the U2AF2. Therefore, we extracted the sequences of the 14 proved alternatively spliced genes from the reference genome (hg19) and uploaded them to RBPmap web server (http://rbpmap.technion.ac.il/) for detecting the existence of the U2AF2-binding motif (provided by CISBP-RNA database, id: M077_0.6) with default parameters. The results in Supplementary Fig. 8a showed that ESRRG contains a significant number of the U2AF2-binding motifs, indicating a great potential to be aberrantly spliced owing to the mutation in SF3B1. Also, the role and function of ESRRG in PRL synthesis and prolactinomas proliferation (mentioned in the main content) attracted our attention to performing detailed analysis on ESRRG.

When splicing, the U2 snRNP complex will interact with the polypyrimidine track with the length of 15~20 bp and located about 5~40 bp before the 3′ end of the intron to be spliced. Under this fact, we extracted the sequences of the last 100 bp at the 3′ end of introns in all annotated transcripts of ESRRG (from RefSeq reference transcriptome, version GRCh37.p13) and further verified the existence of the U2AF2-binding motif on RBPmap web server. The results were shown in Supplementary Fig. 8b and c.

**Differential expression analysis of *ESRRG* on gene- and transcript-level**. Owing to the splicing of ESRRG is potentially related to the mutation in SF3B1, we analyzed the differential expression of ESRRG in two comparison groups: mutant tumor vs normal pituitary and wild-type tumor vs normal pituitary. This setting regarded the normal pituitary as the background and was designed for detecting the variation of ESRRG expression in tumors with/without the SF3B1 mutation. The quantification of transcript-level expression was done by Salmon (https://combine-lab.github.io/salmon/), which is the state-of-the-art tool, with the reference transcriptome from RefSeq (GRCh37.p13), and the differential expression analysis was finished by DESeq2 (https://bioconductor.org/packages/release/bioc/html/DESeq2.html). Both tools were executed by default parameters. The transcript-level results were shown in Supplementary Fig. 8d and e, which led us to recognize the cryptic ESRRG transcript NM_001243518.1. The gene–level analysis can be easily converted by aggregating the transcript-level results, shown as Fig. 3b.

**Digital droplet PCR (ddPCR)**. Detection of rare variants in SF3B1 was performed on the EP1 Digital Array (Fluidigm) Digital PCR system. The SF3B1 mutation analysis is based on allele-specific PCR. We designed external primers for complementary probe regions and TaqMan MGB probes (TsingKe Biological Technology) to detect the mutations in SF3B1 c.1874G > A. One probe targets the mutant variant (tagged with Applied Biosystems' proprietary 5-hexachloro-fluorescein (HEX) fluorophore), and the other targets a wild-type variant (tagged with the 6-carboxyfluorescein (FAM) fluorophore). The TaqMan MGB probes and primers used for ddPCR were listed in Supplementary Table 11.

Two different arrays were used. To test the effects of different PCR components, 1) the 12.765 array (BMK-M10-12.765, Fluidigm) was used with 8 μL reaction mixtures comprising 0.8 μL DNA sample, 0.4 μL 20× GE Sample Loading Reagent (PN 85000746, Fluidigm), 2.8 μL 10 μM gene-specific assays (genotyping primer and TaqMan probes specific for mutated and wild-type SF3B1) and 4 μL TaqMan Gene Expression Master Mix (PN 4369016, Life Technologies); 2) the 48.770 array (dPCR 37k IFCs, 100-6151, Fluidigm) was used with 4 μL reaction mixtures comprising 0.88 μL DNA sample, 0.4 μL 20×GE Sample Loading Reagent, 0.72 μL 20 μM gene-specific assays and 2 μL 2× TaqMan Gene Expression Master Mix. The loaded arrays were then transferred to the EP1 Cycler. Thermocycling was performed as follows: 120 s at 50 °C, a hot start at 95 °C for 10 min, and 40 cycles of 15 s of denaturation at 95 °C, and 1 min of annealing and extension at 60 °C.

EP1 Data Collection and Analysis Software was used to process the data, analyze PCR amplification, and count the numbers of HEX-positive chambers and FAM-positive chambers in each panel. We used 6-carboxy-X-rhodamine signals as an internal positive PCR control. Positive and negative controls were used to assess platform function, amplification protocol and to establish the Cq range and the quantification threshold. A total of 450 bp synthetic DNA fragments for SF3B1 c.1874G > A were used as positive controls (TsingKe Biological Technology). DNA from a healthy individual was used as a negative control.

To determine the specificity of the assay, we performed ddPCR using water and healthy control DNA as a negative control. In control experiments, where no template was added, generally, no positive chambers were detected. Rarely, one to two chambers representing mutant amplifier were detected in healthy control DNA. We set a minimum cutoff frequency of 0.5% (one mutant per 200 total alleles) to call a DNA sample positive for a SF3B1 mutation by ddPCR. Representative ddPCR outputs are shown in Supplementary Table 3. For all samples, the c.1874G > A ddPCRs were performed at least three times.

**Cell culture and adenoviral constructs**. MCF7 cells and rat PA cells (GH3 and MMQ) were originally obtained from the American Type Culture Collection (ATCC) and cultured at 37 °C in 35 mm dishes in a humidified atmosphere of 95% air and 5% $CO_2$. The culture medium of GH3 and MMQ were Ham's F12K medium with 2.5% fetal bovine serum (FBS) and 15% horse bovine serum. The culture medium of MCF7 was Eagle's Minimum Essential Medium with 10% FBS. Cultures were fed every other day. The cell lines were also genotyped to rule out cross-contamination and their morphology was regularly examined.

Prolactin levels were measured using an ELISA kit (K4688, BioVision) according to the manufacture's protocol. We used HEK293 cells as a non-pituitary control. The $SF3B1^{R625H}$ mutant was generated from the human wild-type SF3B1 construct (I1439, obtained from GeneCopoeia), by point mutation using a site-directed mutagenesis kit (210518, QuikChange II, Stratagene). DNA fragments corresponding to full-length (canonical) or cryptic (contains additional 21 bp) ESRRG were amplified from a human cDNA library by PCR and inserted into pDC316-mCMV-ZsGreen expression Vector (Sigma-Aldrich) using the NheI and NotI restriction sites. Adenoviruses expressing each of these constructs were constructed by BAC Biological Technology.

**CCK-8 assay cell growth viability**. Cells after treated or untreated were seeded at a concentration of $4 \times 10^3$ per well in the 96-well plate. Each group was detected with Cell Counting Kit-8 (Beyotime, C0039), following the manufacturer's instructions. In brief, 10 μl CCK-8 were added into each well, and cells were incubated for an additional 4 h. The absorbance at 450 nm was measured using a microplate reader.

**Apoptosis analysis**. Cells were analyzed for apoptosis by an Annexin V-FITC/PI double-staining method described by kit manufacturer (Beyotime, C1062M). The cells 48 h after treatment were collected and subjected to the analysis. About $5 \times 10^5$ cells each group were collected by centrifugation and resuspended 500 μl of binding buffer. 5 μl of Annexin V-FITC and 5 μl of PI were added into each tube, then incubated at room temperature for 15 min in the dark. Stained cells were analyzed by flow cytometry in FITC and PE channels.

**Colony formation assay**. Single-cell suspensions of $1 \times 10^3$ cells were plated in 2 mL of Dulbecco's Modified Eagle Medium (DMEM) containing 10% FBS. During 3 weeks of cell culture, the medium was changed every 3 days. Then the colonies were fixed in 4% paraformaldehyde, and stained with 0.04% crystal violet in PBS for 15 min at room temperature. After extensive washing and air drying, the colony numbers were measured by ImageJ.

**GST pull-down**. In all, 500 μg HIS-cryptic ESRRG protein, 200 μl immobilized GST-tag Purification Resin and 500 μg GST-Pit-1 protein were added to 1000 μl pull-down buffer (50 mM Tris, 150 mM NaCl, 0.1% Triton X-100, 10 mM EDTA, 1 mM PMSF, 1% protease inhibitor cocktail (pH 8.0)), then incubated at 4 °C for 16 h. Similarly, HIS-canonical ESRRG protein was incubated with immobilized GST-Pit-1 protein. As a negative control, HIS-cryptic ESRRG protein and HIS-canonical ESRRG protein was incubated with GST protein. Beads were washed four times with the pull-down buffer. Retained proteins were released by adding 2× loading buffer and boiled for 5 min at 95 °C, then resolved by sodium dodecyl sulfate-polyacrylamide gel electrophoresis (SDS-PAGE) and detected by the GST monoclonal antibody (CUSABIO, CSB-MA000304, 1:200) and His-Tag Monoclonal antibody (CUSABIO, CSB-MA000159, 1:200).

**Immunoprecipitation and immunoblotting**. FLAG-Pit-1 and HA-cryptic ESRRG co-transfected cell extract, 50 μl immobilized protein G Agarose, and 10 μg FLAG antibody (ABclonal, AE005, 1:100) were added to 1000 μl of CoIP buffer (50 mM Tris, 150 mM NaCl, 0.1% Triton X-100, 1 mM PMSF, 1% protease inhibitor cocktail), then incubated at 4 °C for 16 h. Similarly, FLAG-Pit-1 and HA-canonical ESRRG co-transfected cell extract was incubated with FLAG antibody. Human prolactinomas lysates were immunoprecipitated with Pit-1 (Santa Cruz Biotechnology, sc-25258, 1:500) or ESRRG (Abcam, ab16366, 1:100) antibody. As a negative control, cell extract was incubated with mouse IgG. Beads were washed four times with the CoIP buffer. Retained proteins were released by adding 2× loading buffer and boiled for 5 min at 95 °C, then resolved by SDS-PAGE and detected by the Mouse anti-FLAG mAb, anti-HA (Abcam, ab9110, 1:500), anti-ESRRG and anti-Pit-1. GAPDH (Abcam, ab8245, 1:5000) was used as an internal control in western blot.

The simple western immunoblots were performed on a Wes (ProteinSimple) using the Jes/Wes Separation Master Kit (12–230 kDa) according to the manufacturer's standard instruction, using the following antibodies: anti-ESRRG (Abcam, ab49129, 1:400), anti-SF3B1 (Novus, NB100-55255, 1:400) and anti-β-actin (Abcam, ab8227, 1:1000).

**RNA immunoprecipitation (RIP)**. RNA immunoprecipitation was used to investigate whether ESRRG could bind with the potential binding protein SF3B1. We used the EZ-Magna RIP kit (17-701, Millipore) following the manufacturer's protocol. Cells were lysed in complete RIP lysis buffer, and the extract was

incubated with magnetic beads conjugated with antibodies that recognized SF3B1 antibody (sc-514655, Santa Cruz Biotechnology, 1:100) or control IgG (Millipore) for 6 h at 4 °C. Then, the beads were washed and incubated with Proteinase K to remove proteins. Finally, purified RNA was subjected to qRT-PCR analysis to demonstrate the presence of ESRRG uses specific primers. Primer pairs for RIP were listed in Supplementary Table 9.

**UV CLIP and qPCR**. The CLIP assay was adapted from the previous publications[22]. Specifically, MCF cells were washed in ice-cold PBS and PBS was removed completely. Plates were placed in a UV crosslinker and irradiated with 150 mJ cm$^{-2}$ of UVA (365 nm) before being lysed. After lysis, cell lysates were incubated with RNase T1 (ThermoFisher Scientific, EN0541) at 1 U μl$^{-1}$ at 22 °C for 6 min to digest RNAs that were not protected from bound proteins, then subjected to immunoprecipitation with an anti-SF3B1 (sc-514655, Santa Cruz Biotechnology) or IgG antibody following standard RIP protocol[22]. The immunoprecipitated RNA was isolated using the PureLink RNA Mini Kit (ThermoFisher Scientific, 12183018 A) with DNase treatment. After reverse transcription (Takara, 638313), the resultant cDNA was subjected to qRT-PCR assay. The primers used for qRT-PCR were designed to cover the full-length human ESRRG sequence and listed in the Supplementary Table 8. Data are normalized to IgG (SF3B1 IP/IgG IP).

**Culture of primary human prolactinoma cells**. Human prolactinomas were obtained at the time of surgery and transferred in fresh L15 medium enriched with 10% FBS. The PAs tissues were cut into small pieces and then were digested with collagenase (1 mg ml$^{-1}$; 17101015, Thermo Fisher) for 30 min at 37 °C. After terminating the enzymatic treatment by addition of FBS, the mixture was filtered with cell strainer to remove undigested tissues and centrifuged at 600 × g for 5 min. The cell pellet was resuspended in Neurobasal growth medium supplemented with 2% B27 (A3582801, Thermo Fisher) and plated on 35 mm dishes. Tumor cells were infected with adenovirus at MOI of 30 or 100 for 48 h. Tumor cells were digested and centrifuged for 5 minutes and suspended in the medium. Live cells were calculated and re-plated in 24-well plates. The supernatant was collected for analysis for prolactin secretion after 24-hour culture. Prolactin levels were measured using an enzyme-linked immunosorbent assay (ELISA) (CSB-E06883h, Cusabio) according to the manufacture's protocol.

**Transfection and RNA interference**. siRNA transfections were performed using Lipofectamine 2000 (11668019, Thermo Fisher), according to the manufacturer's protocol. siRNA synthesis was performed by Shanghai GenePharma and the siRNA sequences for human SF3B1 are shown in Supplementary Table 9.

**QRT-PCR**. Total RNA was extracted using RNeasy Mini Kit (74104, Qiagen) and then reversed transcribed using High Capacity cDNA Reverse Transcription Kit (4368814, Thermo Fisher) according to the manufacturer's instructions. Subsequently, we performed qRT-PCR using Power SYBR Green PCR Master Mix (4367659, Thermo Fisher) in a total reaction volume of 10 μL. GAPDH was used as a reference gene. The levels of mRNAs were performed on an ABI 7500 System (Applied Biosystems). Primer pairs for qRT-PCR are shown in Supplementary Table 9. Amplification was performed as follows: 95 °C for 10 min and 40 cycles at 95 °C for 15 sec, 60 °C for 60 sec. For the quantitative analysis, relative expression levels were calculated based on CT values (corrected for GAPDH expression) according to the equation: $2^{-\Delta CT}$ [ΔCT = CT (gene of interest) – CT (GAPDH)]. All qRT-PCR analyses were performed in triplicate.

Total RNA was extracted using RNeasy Mini Kit (74104, Qiagen), and reverse transcription was performed using High Capacity cDNA Reverse Transcription Kit (4368814, Thermo Fisher) according to the manufacturer's protocol. Subsequently, we performed PCR using I-5 High-Fidelity Master Mix (I5HM - 200, MCLAB). The thermocycling protocol was listed as follows: initial denaturation at 98 °C for 2 min, followed by 32 repeats of the three-step cycling program consisting of 10 sec at 98 °C (denaturation), 10 sec at 59 °C (primer annealing) and 10 sec at 72 °C (elongation), followed by a final extension step for 5 min at 72 °C. Primers were designed to obtain an amplification product that spans the Cufflinks-predicted alternative splicing junctions. PCR reactions were carried out using primers/ conditions described in Supplementary Table 10, and PCR products were run on 1–3% agarose gel and visualized using a UV transilluminator.

**Luciferase reporter assays**. A reporter gene containing an upstream fragment of 2 kilobases of the 5′ promoter region in the human PRL gene linked to pGL3-basic reporter vector. ESRRG-cryptic (NM_001243518.1), ESRRG-canonical (NM_001243519.1) cDNA were, respectively, cloned into pCDH vector. Human Pit-1 cDNA (NM_000306.1) was respectively cloned into pCDNA3.1 vector. All constructs were verified by Sanger sequencing (TsingKe Biological Technology). HEK293 cells were cultured at 37 °C in a humidified atmosphere of 95% air and 5% CO$_2$. The culture medium was DMEM with 10% FBS. Cultures were fed every other day. The cell lines were also genotyped to rule out cross-contamination and their morphology was regularly examined. HEK293 cells were transfected in 24-well plates containing 2 μL of Lipofectamine 2000, 1 μg of pCDH-ESRRG-cryptic/ pCDH-ESRRG-canonical and/or pCDNA3.1/ pCDNA3.1-Pit-1, 1 μg of pGL3-basic

or pGL3-basic-pRL-p, and 100 ng of pRL-TK-Renilla (as transfection control) for 48 h. The cells were lysed in buffer (100 μL lysis buffer, Promega Corporation) and luciferase activity was then measured in a Mithras LB 940 apparatus (Berthold Technologies). The different vectors transfection groups are shown in Supplementary Table 12.

**Statistical analysis**. The statistical analysis was performed using SPSS version 22.0 (SPSS Inc.) and graphs were prepared with Prism 7.0 (GraphPad) software. T test or Mann–Whitney U test were performed for comparison of continuous variables between two groups. Comparison of three or more groups was performed with one-way analysis of variance followed by Tukey's multiple comparisons post hoc test, Dunnett's multiple comparisons test or Bonferroni multiple comparisons test. Pearson's $\chi^2$ test was performed for comparison of categorical variables between two groups. All the experiments were performed in triplicates. $P < 0.05$ was considered to indicate statistical significance.

**Reporting summary**. Further information on research design is available in the Nature Research Reporting Summary linked to this article.

## Data availability

The raw sequence data reported in this paper have been deposited in the Genome Sequence Archive in BIG Data Center, Beijing Institute of Genomics (BIG), Chinese Academy of Sciences, under accession numbers HRA000041, HRA000041 that can be accessed at https://bigd.big.ac.cn/gsa-human/browse/HRA000041. The deposited and publicly available data are compliant with the regulations of the Ministry of Science and Technology of the People's Republic of China. The raw sequencing data and somatic and germ-line mutation calls contain information unique to an individual, require controlled access, all data are available from the corresponding author upon reasonable request. The source data underlying Table 1, Figs. 1, 2, 3a, b, d, e, g, 5a–e, h, 6, 7a, b and Supplementary Figs 3a, 6a–c, 8d, and 11 are provided as a Source Data file.

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

## Acknowledgements

We thank Mrs. Wei Qi, Mrs. Yue Wang, Mr. Ce Ma, and the Novogene Biotechnology Inc for technical support. We also thank Mr. Qiuyue Fang for their technical assistance with western blot. This work was supported by the Research Special Fund for Public Welfare Industry of Health (201402008), the National High Technology Research and Development Program of China (863 Program, 2015AA020504), Beijing Municipal Science & Technology Commission (Z171100000117002), and National Natural Science Foundation of China (81672495).

## Author contributions

Y.Z. and Z.Z. conceived the project, C.L. and W.X. designed the experiments, analyzed the data, and wrote the manuscript. J.R. helped to revise the manuscript. J.G., Y.S., H.W., L.G., M.L., S.Z., and S.C. performed the experiments. J.Z. and T.J. performed the alternative splicing analysis. Y.M., H.Z.(Haibo Zhu), H.Z.(Hongwei Zhang), M.Z., Y.Q., S.L., and F.W. assisted with the management of clinical data and specimens. G.L. and J.W. assisted with the pathological assessment of biospecimens. J.M. assisted with the neuroradiological assessment. Q.Z. helped to polish the manuscript. All authors read and approved the manuscript.

## Competing interests

The authors declare no competing interests.
