## [Peer Review File · Nature Communications]

Reviewers' comments:

Reviewer #1 (Remarks to the Author):

This is a very impressive manuscript combining molecular biology with large clinical database, which addresses the role of the somatic SF3B1 hotspot mutation in prolactinomas. It is a very comprehensive study with interesting findings. In this manuscript, Li et al. investigate the role of SF3B1 hotspot mutation in prolactinomas. By the whole genome sequencing and digital polymerase-chain-reaction (PCR) analysis, they showed the hotspot somatic mutation in splicing factor 3 subunit B1 (SF3B1R625H) in 19.8% prolactinomas for the first time. Mechanism study reveals that SF3B1R625H mutation causes aberrant splicing of ESRRG and leads to stronger binding with Pit-1. This physiological process leads to increased cell proliferation and prolactin secretion. Overall, this manuscript highlights the importance of SF3B1R625H and it may be a potential therapeutic target of prolactinomas.

Major concerns:

1. This study includes a large number of cases of prolactinoma. According to Table S5 "Inclusion and exclusion criteria", one of the criteria is that prolactin is greater than 90 ng/ml. Normally, we diagnose prolactinoma by requiring prolactin greater than 150 or 200 ng/ml. Author should clarify this. Obviously, symptoms of prolactinoma should also be included as one of the criteria. In addition, it is well known that the first choice for prolactinomas is dopamine agonists. Thus, the majority of cases are resistant to DA? There is a need for clarity in the article.
2. The author found that SF3B1 mutation resulted in a significant augment of PRL secretion in human prolactinoma cells. Does the PRL level significantly elevate in this series of mutant prolactinomas from the table 1? Why does the author choose the clinical characteristic of PRL (ng/ml)/Tumor Size (cm) instead of PRL? Line 210-213, "Patients with mutant SF3B1 showed higher levels of PRL (plasma PRL/tumor size) than the wild type (WT) group ($p=0.02$, Mann-Whitney U-test, Fig. 5b, Table 1), indicating that SF3B1 mutant prolactinomas have higher PRL production. The expression of this sentence does not seem to be appropriate.
3. In Fig.3, the author revealed that SF3B1R625H mutation significantly promoted cell proliferation, but in Table 1, SF3B1R625H mutation dose not positively correlated with tumor size ($p= 0.74$). Is it appropriate that the maximal tumor diameter is used to represent the tumor volume?
4. The co-IP experiments in Fig. 4 are performed with over-expressed tag-proteins in HEK293 cells; do the authors have any data using antibodies against the native proteins and in normal physiological conditions (rather than with over-expressed proteins)?
5. I believe that a number of prolactinomas still need medication after surgery to control prolactin levels, especially in invasive cases. So the result of PFS is related to the drug response? In the absence of such data, the author should explain the limitations of the study in the Discussion.

Minor concerns:

1. All of knockdown experiments should be performed with two individual siRNAs.
2. Line 43, what is "LAs"?
3. In the section of Introduction, the author should introduce the reason and/or background for this study.
4. Line 130, what is the "human SSC line"?
5. Table 1, how does the author define the "Tumor invasion", which should be illustrated?
6. In Fig. 2d and g, the author wants to show that overexpression of SF3B1R625H would improve the colony formation of GH3 and MMQ cells. Please provide the original photograph of these experiments. Additionally, add more colony formation experiments of SF3B1 knockdown or knockout.
7. Line 197, is this Fig. 3b?
8. Two prolactinomas which displayed unusual malignancy have SF3B1R625H mutant?
9. The author should provide all of the detailed experiments methods, such as CCK-8, colony formation, GST-pulldown, immunoprecipitation and the methods of primary cell culture to Methods section.

Reviewer #2 (Remarks to the Author):

In this study, Li et. al. reported somatic SF3B1 hotspot mutations in prolactinomas. Through whole-genome sequencing analysis and PCR assays, they found that SF3B1R625H is a recurrent mutation in prolactinomas. Patients with this mutation expressed higher levels of prolactin and showed poor prognosis. They also wished to conclude that SF3B1R625H promotes aberrant splicing of the estrogen-related receptor gamma, whose product binds to the pituitary-specific transcription factor and stimulates prolactin production. While identification of frequent mutations in prolactinomas is important, the rigor of current study requires substantial improvement in order for consideration of publication in Nature Communications or other peer reviewed journals. A few points that dampened the reviewer's enthusiasm are listed below.

The scientific rigor needs to be substantially improved. In many instances, the information describing the results was absent. It is difficult to judge whether the RNA sequencing analysis was done properly as no descriptions on the samples that were used to compare and analyze. It is also unclear how the RBPmap analysis was performed. In the siRNA knockdown experiment, there was no western blot analysis to show the SF3B1siRNA knockdown efficiency. Fig. S5h, the sizes of the splicing products are incorrect.

In some instances, the data reported by the authors were not consistent. On page 5, line 95-102, the authors indicated that the total PA sample number was 154 including 120 non-prolactinomas and 34 prolactinomas. This would indicate that prolactinomas is ~25% of the total PAs, a result that is inconsistent with the authors description that prolactinomas is comprised of half of the PAs. The authors also showed that 2 out of 34 prolactinomas harbored the SF3B1 mutation - this was significantly lower than the nearly 20% occurrence the authors described in this study.

What defines cryptic and canonical splicing? It is unclear why the authors call cryptic ESRRG splicing, as from the Genome Browser, the "cryptic ESRRG splicing" is an alternatively spliced variant. Moreover, in order to conclude that SF3B1R625H regulates ESRRG alternative splicing, a splicing minigene reporter should be used for analysis.

Reviewer #3 (Remarks to the Author):

Through genomic analysis of a large number of clinical samples, the authors identified SF3B1 R625H hotspot mutation in 20% of prolactinomas. SF3B1 is a component of the U2 small nuclear ribonucleoproteins complex that participates in the pre-mRNAs splicing. Mutations of SF3B1 have been reported in myelodysplastic syndrome (MDS), chronic lymphocytic leukemia (CLL), and uveal melanoma with observed alternative splicing events. The authors performed RNA seq and identified aberrant spliced transcripts in SF3B1 mutant cells. Among them, they showed that aberrant splicing of estrogen related receptor gamma (ESRRG) could result in stronger binding with pituitary-specific positive transcription factor 1 (Pit-1), leading to excessive PRL secretion. Importantly, they demonstrated that SF3B1 is a worse prognostic marker. The genomic data are convincing, the mechanistic link of the mutation to PRL production has been tested by a series of biochemical and genetic data. The research is both biologically and clinically important to the most common primary CNS tumor. A few concerns need to be addressed

1. Fig 4. Please clarify that the ESRRG siRNAs were specifically designed for the cryptic ESRRG (the longer version), which is the driver of PRL overexpression. If not, a cryptic ESRRG specific siRNA can be designed and the experiments can be done.
2. The siRNA knockdown effects need to be shown by western blotting.
3. Table 2 can be moved to supplement

4. Fig 1B is very busy. The two validation cohorts can be combined together and become Fig 1C.
5. Methods of WGS, RNA seq, Microfluidic-chamber-based digital PCR, primary cell cultures, rMATS analysis are missing.

Reviewers' comments:

Reviewer #1:

This is a very impressive manuscript combining molecular biology with large clinical database, which addresses the role of the somatic SF3B1 hotspot mutation in prolactinomas. It is a very comprehensive study with interesting findings. In this manuscript, Li et al. investigate the role of SF3B1 hotspot mutation in prolactinomas. By the whole genome sequencing and digital polymerase-chain-reaction (PCR) analysis, they showed the hotspot somatic mutation in splicing factor 3 subunit B1 (SF3B1R625H) in 19.8% prolactinomas for the first time. Mechanism study reveals that SF3B1R625H mutation causes aberrant splicing of ESRRG and leads to stronger binding with Pit-1. This physiological process leads to increased cell proliferation and prolactin secretion. Overall, this manuscript highlights the importance of SF3B1R625H and it may be a potential therapeutic target of prolactinomas.

Major concerns:

1) Comment: This study includes a large number of cases of prolactinoma. According to Table S5 “Inclusion and exclusion criteria”, one of the criteria is that prolactin is greater than 90 ng/ml. Normally, we diagnose prolactinoma by requiring prolactin greater than 150 or 200 ng/ml. Author should clarify this. Obviously, symptoms of prolactinoma should also be included as one of the criteria. In addition, it is well known that the first choice for prolactinomas is dopamine agonists. Thus, the majority of cases are resistant to DA? There is a need for clarity in the article.

Reply: We appreciate your thoughtful suggestion. We agree with the reviewer that PRL levels of greater than 150-200 ng/ml are used routinely in the diagnosis of prolactinomas. However,

according to Niki Karavitaki, it is unlikely that nonfunctionary macroadenomas present with serum PRL > 94.31 ng/ml¹. Therefore, in our study, we felt it appropriate to make the diagnosis of prolactinoma with serum PRL > 90 ng/ml and confirmed positive PRL IHC staining.

Symptoms of prolactinoma should also be included as one of the entry criteria and almost all patients have varying degrees of symptoms related to the tumors.

In our study, reasons for surgical intervention did include first line therapy (dopamine agonist) resistance as well as sudden exacerbation of clinical symptoms, including visual impairment, tumor stroke and patients intolerance of side effects of drugs or reluctance to take drugs for a long time.

2) Comment: The author found that SF3B1 mutation resulted in a significant augment of PRL secretion in human prolactinoma cells. Does the PRL level significantly elevate in this series of mutant prolactinomas from the table 1? Why does the author choose the clinical characteristic of PRL (ng/ml)/Tumor Size (cm) instead of PRL? Line 210-213, “Patients with mutant SF3B1 showed higher levels of PRL (plasma PRL/tumor size) than the wild type (WT) group (p=0.02, Mann-Whitney U-test, Fig. 5b, Table 1), indicating that SF3B1 mutant prolactinomas have higher PRL production.” The expression of this sentence does not seem to be appropriate.

Reply: Thank you for your suggestion. First, the difference of plasma PRL level between SF3B1 mutation and wild group is significant (Mann Whitney test, p=0.0462). But we think that PRL secretion per unit tumor volume is more responsive to the secretory capacity of tumor cells, because PRL levels generally correlate with prolactinoma size^{2,3}. Therefore we

choose the PRL (ng/ml)/Tumor Size (cm) instead of PRL. And there have been similar report before⁴. But there is a difference between “levels of PRL” and “plasma PRL/tumor size” and we can't use them interchangeably. We have changed accordingly.

3) Comment: In Fig.3, the author revealed that SF3B1R625H mutation significantly promoted cell proliferation, but in Table 1, SF3B1R625H mutation dose not positively correlated with tumor size (p= 0.74). Is it appropriate that the maximal tumor diameter is used to represent the tumor volume?

Reply: We appreciate your thoughtful suggestion. There are many ways to calculate the volume of tumors. The maximum diameter and volume formulas ($\pi/6 \times \text{length} \times \text{width} \times \text{height}$) may be used. Both methods of calculation show that *SF3B1*^{R625H} mutation does not positively correlate with tumor size. As far as tumor volume and cell proliferation are concerned, we think that tumor growth in vivo is very complex and tumor volume is affected by many confounding factors, so cell proliferation experiments in vitro are not directly related to tumor volume.

4) Comment: The co-IP experiments in Fig. 4 are performed with over-expressed tag-proteins in HEK293 cells; do the authors have any data using antibodies against the native proteins and in normal physiological conditions (rather than with over-expressed proteins)?

Reply: Following the thoughtful suggestions from the reviewers, we have supplemented the native co-IP experiments in the pituitary adenoma tissue samples according to the reviewer's suggestion, The results indicating a physical interaction between Pit-1 and ESRRG in human pituitary adenoma), details please refer to Supplementary Figure 11.

5) Comment: I believe that a number of prolactinomas still need medication after surgery to control prolactin levels, especially in invasive cases. So the result of PFS is related to the drug response? In the absence of such data, the author should explain the limitations of the study in the Discussion.

Reply: We appreciate your thoughtful suggestions. All patients in the study group were treated with a unified treatment regimen. Postoperative dopamine agonist was used to controlling hyperprolactinemia. The problem of dopamine resistance may be an influential factor for PFS. We have changed according to your advice from line 268 to 270.

Minor concerns:

6) Comment: All of knock down experiments should be performed with two individual siRNAs.

Reply: We agree with the reviewer's comments and supplemented the corresponding results in the article Fig 2b.

7) Comment: Line 43, what is "LAs"?

Reply: We appreciate for your comments, we have corrected this with "prolactinomas".

8) Comment: In the section of Introduction, the author should introduce the reason and/or background for this study.

Reply: We have added the relevant content according to the requirements of reviewer, details please refer to line 54-75.

9) Comment: Line 130, what is the "human SSC line"?

Reply: We are sorry for the clerical error and have deleted these words.

10) Comment: Table 1, how does the author define the "Tumor invasion", which should be

illustrated?

Reply: We appreciate your thoughtful suggestions. The diagnostic criteria for tumor invasion include image Knosp grading⁵ (Grade 3 and 4), intraoperative or pathological evidence of tumors invading dura, skull bone and brain tissue.

11) Comment: In Fig. 2d and g, the author wants to show that overexpression of SF3B1R625H would improve the colony formation of GH3 and MMQ cells. Please provide the original photograph of these experiments. Additionally, add more colony formation experiments of SF3B1 knockdown or knockout.

Reply: We have added the relevant content according to the requirements of reviewer, details please refer to Supplementary Figure 4.

12) Comment: Line 197, is this Fig. 3b?

Reply: Yes, it should be Fig.4f here, we have corrected this.

13) Comment: Two prolactinomas which displayed unusual malignancy have SF3B1R625H mutant?

Reply: We appreciate your thoughtful suggestions. The two prolactinomas displayed unusual malignancy, details please refer to line 233-237.

14) Comment: The author should provide all of the detailed experiments methods, such as CCK-8, colony formation, GST-pulldown, immunoprecipitation and the methods of primary cell culture to Methods section.

Reply: In the previous version, we put most of the method descriptions in the supplementary material, now we put the major methods into the MATERIALS AND METHODS section of the manuscript.

Reviewer #2:

In this study, Li et. al. reported somatic SF3B1 hotspot mutations in prolactinomas. Through whole-genome sequencing analysis and PCR assays, they found that SF3B1R625H is a recurrent mutation in prolactinomas. Patients with this mutation expressed higher levels of prolactin and showed poor prognosis. They also wished to conclude that SF3B1R625H promotes aberrant splicing of the estrogen-related receptor gamma, whose product binds to the pituitary-specific transcription factor and stimulates prolactin production. While identification of frequent mutations in prolactinomas is important, the rigor of current study requires substantial improvement in order for consideration of publication in Nature Communications or other peer reviewed journals. A few points that dampened the reviewer's enthusiasm are listed below.

1) Comment: The scientific rigor needs to be substantially improved. In many instances, the information describing the results was absent.

Reply: Thanks for your suggestion. We have some changes to the manuscript to make it more logical and clear.

2) Comment: It is difficult to judge whether the RNA sequencing analysis was done properly as no descriptions on the samples that were used to compare and analyze. it is also unclear how the RBPmap analysis was performed.

Reply: Thanks for your suggestion. Based on the reviewer's comment, we added some analysis methods details, make the article more scientific rigor, details please refer to the MATERIALS AND METHODS section (RNA sequencing, Alternative splicing analysis,

Differential expression analysis of ESRRG on gene- and transcript-level).

3) Comment: In the siRNA knockdown experiment, there was no western blot analysis to show the SF3B1 siRNA knockdown efficiency.

Reply: Based on the reviewer's comments, we performed western blot experiments to show the SF3B1 siRNA knockdown efficiency (Supplementary Figure 3b, 9).

4) Comment: Fig. S5h, the sizes of the splicing products are incorrect.

Reply: We appreciate your thoughtful suggestions. In our original version of supplementary material, there is no Fig. S5h, it refers to the other figure error? I would like to ask whether the present version of Fig.S5h (the original version of Fig.S4 has the errors of the splicing products)? --Canonical and cryptic products sizes are 176 and 170bp, respectively, so the two bands appear very close together on the graph.

5) Comment: In some instances, the data reported by the authors were not consistent. On page 5, lane 95-102, the authors indicated that the total PA sample number was 154 including 120 non-prolactinomas and 34 prolactinomas. This would indicate that prolactinomas is ~25% of the total PAs, a result that is inconsistent with the authors description that prolactinomas is comprised of half of the PAs. The authors also showed that 2 out of 34 prolactinomas harbored the SF3B1 mutation - this was significantly lower than the nearly 20% occurrence the authors described in this study.

Reply: We appreciate your thoughtful suggestions. Dopamine agonist treatment is the first choice for prolactinomas, so the proportion of surgical cases is gradually decreasing. Another

reason is that the selection of validated cases is not a retrospective cohort and may do not reflect the true proportion of tumors. In the new Supplementary Table 1, we have explained the pathological types of enrolled cases.

In 154 cases of non-prolactinoma screening set, 18 thyrotroph adenoma, 33 somatotroph, 30 gonadotroph, 24 corticotroph, 15 null cell, 16 plurihormonal and 18 ammosomatotroph/mixed adenomas. In this group, only 2 tumors have the mutation. These two tumors with the mutation have a positive immunostaining for PRL. Thus, the *SF3B1*^{R625H} mutation was only detected in PRL immune-positive PAs. And the overall mutation ratio in prolactinomas is about 20% (45/227, 19.8%).

Supplementary Table 1 Overview on the patients included in this study and summary of SF3B1 mutational status of these patients.

6) Comment: What defines cryptic and canonical splicing? It is unclear why the authors call cryptic ESRRG splicing, as from the Genome Browser, the “cryptic ESRRG splicing” is an alternatively spliced variant. Moreover, in order to conclude that SF3B1R625H regulates ESRRG alternative splicing, a splicing minigene reporter should be used for analysis.

Reply: We appreciate your thoughtful suggestions. (1) The reference⁶ analyzed the cryptic splicing event by setting a cutoff value for the results of rMATS. So, we use a similar criterion to define the cryptic transcripts: we collected data sets from the rMATS output on the basis of FDR <0.05. Then we selected top 20 significant aberrant events (IncLevelDifference > 0.3 or IncLevelDifference < -0.3) and the involved gene targets in mutant and wild type tumors using the rMATS pipeline and further validated them using RT-PCR. We define it as cryptic transcripts that dominant transcriptional isoform in mutants, similarly as canonical transcripts in wild type. (2) To confirm the regulation of SF3B1^{R625H} on ESRRG, we infected the adenovirus carrying SF3B1^{R625H} mutation in primary cultured tumor cells, and the RT-PCR results showed that the cryptic ESRRG transcript was only observed in the Ad-SF3B1^{R625H} group (Fig. 3g). The same results were also observed in human MCF7 cell lines (Supplementary Figure 8), Sanger sequence analysis showed that these were the expected fragment with 21 bp elongation of exon 5 (Supplementary Figure 8b, c). We also did minigene reporter analysis to confirm this result. After infection of adenovirus expressing SF3B1^{WT} and SF3B1^{R625H} respectively, this reporter was spliced to produce two major products when co-transfected into MCF7 cells: a fully spliced RNA containing exons 5 (Supplementary Figure 8e, left and middle lanes), and a larger transcript that retained 21bp (Supplementary Figure 8e, right lane).

Reviewer #3:

Through genomic analysis of a large number of clinical samples, the authors identified SF3B1 R625H hotspot mutation in 20% of prolactinomas. SF3B1 is a component of the U2 small nuclear ribonucleoproteins complex that participates in the pre-mRNAs splicing. Mutations of

SF3B1 have been reported in myelodysplastic syndrome (MDS), chronic lymphocytic leukemia (CLL), and uveal melanoma with observed alternative splicing events. The authors performed RNA seq and identified aberrant spliced transcripts in SF3B1 mutant cells. Among them, they showed that aberrant splicing of estrogen related receptor gamma (ESRRG) could results in stronger binding with pituitary-specific positive transcription factor 1(Pit-1), leading to excessive PRL secretion. Importantly, they demonstrated that SF3B1 is a worse prognostic marker. The genomic data are convincing, the mechanistic link of the mutation to PRL production has been tested by a series of biochemical and genetic data. The research is both biologically and clinically important to the most common primary CNS tumor. A few concerns need to be addressed

1) Comment: Fig 4. Please clarify that the ESRRG siRNAs were specific designed for the cryptic ESRRG (the longer version), which is the driver of PRL overexpression. If not, a cryptic ESRRG specific siRNA can be designed and the experiments can be done.

Reply: We totally agree with the reviewer, but compared with the canonical ESRRG, cryptic ESRRG only in exon5 extended the 21 bp, according to the 21 bp, we can only designed one cryptic ESRRG specific siRNA, but ESRRG knockdown effect is not obvious. To confirm the effect of cryptic ESRRG on prolactin secretion, we infected the adenovirus carrying cryptic ESRRG in primary cells, and the results showed that the cryptic ESRRG could promote the secretion of PRL in primary cells, as shown in Supplementary Figure 11.

2) Comment: The siRNA knockdown effects need to be shown by western blotting.

Reply: Thanks for your suggestion, we performed western blot experiments to show the SF3B1 siRNA knockdown efficiency (Supplementary Figure 3b, 9).

3) Comment: Table 2 can be moved to supplement.

Reply: Thanks for your suggestion. We have transferred Table 2 to supplementary materials (Supplementary Table 1) according to your requirements.

4) Comment: Fig 1B is very busy. The two validation cohorts can be combined together and become Fig 1C.

Reply: Thanks for your suggestion. We have already revised it according to your requirements.

5) Comment: Methods of WGS, RNA seq, Microfluidic-chamber-based digital PCR, primary cell cultures, rMATS analysis are missing.

Reply: Thanks for your suggestion. In the previous version, we put most of the method descriptions in the supplementary material, now we put the major methods into the MATERIALS AND METHODS section of the manuscript.

Reference

1. Karavitaki N, *et al.* Do the limits of serum prolactin in disconnection hyperprolactinaemia need re-definition? A study of 226 patients with histologically verified non-functioning pituitary macroadenoma. *Clin Endocrinol (Oxf)* **65**, 524-529 (2006).
2. Melmed S, *et al.* Diagnosis and treatment of hyperprolactinemia: an Endocrine Society clinical practice guideline. *J Clin Endocrinol Metab* **96**, 273-288 (2011).
3. Casanueva FF, *et al.* Guidelines of the Pituitary Society for the diagnosis and management of prolactinomas. *Clin Endocrinol (Oxf)* **65**, 265-273 (2006).
4. Ma ZY, *et al.* Recurrent gain-of-function USP8 mutations in Cushing's disease. *Cell Res* **25**, 306-317 (2015).
5. Knosp E, Steiner E, Kitz K, Matula C. Pituitary adenomas with invasion of the cavernous sinus space: a magnetic resonance imaging classification compared with surgical findings. *Neurosurgery* **33**, 610-617; discussion 617-618 (1993).

6. Dolatshad H, *et al.* Cryptic splicing events in the iron transporter ABCB7 and other key target genes in SF3B1-mutant myelodysplastic syndromes. *Leukemia* **30**, 2322-2331 (2016).

Reviewers' comments:

Reviewer #1 (Remarks to the Author):

none

Reviewer #2 (Remarks to the Author):

The revised manuscript has enhanced its quality. While the results are interesting, there still remain many problems regarding the rigor of the experiments and the conclusions that were not supported by the presented data. Specific comments are:

1. Fig. 2e, 2h: There is a minute difference in cell proliferation when expressing SF3B1R625H, compared to control SF3B KD cells, and this difference was either not observed between SF3B1R625H and WT or the change was extremely minor. Similar cases were shown in Fig. 2d, 2g. These results do not support the claim that "Overexpression of SF3B1R625H by adenovirus markedly improved the focus formation of GH3 and MMQ cells" and that "SF3B1R625H increased the growth of GH3 and MMQ cells".

The labels in Fig. 2m were unclear, making it impossible to judge the data.

With these problems, the effect of SF3B1R625H on cell proliferation and cell death was inconclusive. Furthermore, the effects of WT ESRRG and cryptic ESRRG on proliferation and cell death should be shown.

2. Fig. S5a. The authors plotted the composition of top 112 AS events in WT cells and a different set of top 112 AS events in the SF3B1R625H cells. This comparison does not make sense. The authors should delete the WT column.

3. The binding of U2AF should be focused on the vicinity of the alternative exons instead of the entire gene body of the differentially spliced genes. No evidence in Fig. S7 support the claim that "This suggested the impact of SF3B1R625H on the splicing of ESRRG through association with the U2 snRNP."

4. Fig. 3b. Cross-linking and Immunoprecipitation (CLIP) is a golden standard to determine the binding of an RNA binding protein on RNA. RIP signals can be caused by indirect interactions between proteins and RNA. Regardless, no SF3B1 mutant was included in this RIP experiment in Fig. 3b, and no description on the location of the primers for PCR. The PCR primers should be located at the vicinity of the branch point of the A3SS sites. It remains unclear whether the SF3B1 mutant has a higher binding affinity at the vicinity of the A3SS site of the ESRRG.

5. Fig. 3c. The authors show that the expression levels of ESRRG is significantly higher in the SF3B1R625H cells than that in the WT cells. This suggests that the SF3B1R625H mutant promotes ESRRG transcription in addition to A3'SS. The description of the results was confusion.

6. Fig. 4b. The authors showed in Fig. 3b that there was a 3-fold increase in WT SF3B1 binding to ESRRG. However, this enrichment was not observed in Fig. 4b (see left two bars, also note the label, it should be IgG and SF3B1). How to explain this discrepancy? Note that the reviewer's comment is based on current figures. PCR primers need to be located in the vicinity of the A3'SS. Together, the conclusion that "This confirmed that the mutant SF3B1 would trigger the aberrant splicing in ESRRG." Is not valid.

7. Fig. S10. Western blot of WT and cryptic ESRRG expression needs to be shown to make sure that the expression levels of WT ESRRG and the cryptic ESRRG are comparable.

8. Fig. 4d. Results showed that KD of ESRRG caused a roughly 2-fold decrease in PRL level. Should it be expected that ectopically expressing WT ESRRG should rescue the decreased PRL expression, which was not the case seen in Fig. S10.

9. SF3B1R625H was previously identified as a prevalent SF3B1 mutant in a few published papers. The authors should acknowledge previous findings when they first describe SF3B1R625H in line 109-111. Examples are:

- Recurrent mutations at codon 625 of the splicing factor SF3B1 in uveal melanoma. Nat Genet. 2013 Feb; 45(2): 133–135.
- Cancer-associated SF3B1 mutations affect alternative splicing by promoting alternative branchpoint usage. Nature Communications volume 7, Article number: 10615 (2016)
- Recurrent hotspot SF3B1 mutations at codon 625 in vulvovaginal mucosal melanoma identified in a study of 27 Australian mucosal melanomas. Oncotarget. 2019; 10:930-941.

Added by Ed

Reviewer #3 (Remarks to the Author):

(none provided)

Reviewers' comments:

Reviewer #2:

The revised manuscript has enhanced its quality. While the results are interesting, there still remain many problems regarding the rigor of the experiments and the conclusions that were not supported by the presented data. Specific comments are:

1) **Comment 1A:** Fig. 2e, 2h: There is a minute difference in cell proliferation when expressing SF3B1R625H, compared to control SF3B KD cells, and this difference was either not observed between SF3B1R625H and WT or the change was extremely minor. Similar cases were shown in Fig. 2d, 2g. These results do not support the claim that “Overexpression of SF3B1R625H by adenovirus markedly improved the focus formation of GH3 and MMQ cells” and that “SF3B1R625H increased the growth of GH3 and MMQ cells”.

Author Response: Thank you. First, the reviewer mentions the comparison between the mutant transfected cell and the control SF3B KD cells. We would like to clarify that in this experiment, the control was a cell transfected with null empty vector virus, not SF3B1 knockdown (KD) cells. What we have been emphasizing in the text is the difference between the SF3B1^{R625H} group (rather than the SF3B1^{WT} group) and the control (Ad-null) group. Second, the gene-transfected cell does show changes, even though minute, that is statistically significant. We speculated that the difference was relatively small in previous results, possibly because the infection efficiency of SF3B1 was not high enough. In order to improve the expression efficiency of SF3B1, we infected GH3/MMQ cells with Ad-SF3B1 at multiplicity of infection (MOI) 30 or 100, the representative western blot analysis of SF3B1 are shown in Supplementary Figure 4. The results showed that SF3B1^{R625H} significantly promoted cell

proliferation in the MOI 100 group compared with control group (Fig. 2d, e, g, h).

Changes to Text:

Line 124-128: We then investigated the role and function of *SF3B1* R625H mutation in the development of prolactinomas by performing colony formation, cell counting kit-8 (CCK-8) and flow cytometry on GH3/MMQ rat pituitary cells with ectopic gene expression as indicated multiplicity of infection (MOI) (Supplementary Figure 4) and stable cell line, respectively.

Comment 1B: The labels in Fig. 2m were unclear, making it impossible to judge the data.

Author Response: We have replaced the new figure (Fig.2m) according to the requirements of the reviewer.

Comment 1C: With these problems, the effect of SF3B1R625H on cell proliferation and cell death was inconclusive. Furthermore, the effects of WT ESRRG and cryptic ESRRG on proliferation and cell death should be shown.

Author Response: We have updated the experimental results of SF3B1 R625H mutation on the apoptosis and proliferation of pituitary tumor cells (Fig. 2d, e, g, h). As for the mechanism of canonical and cryptic ESRRG on proliferation and apoptosis of pituitary tumor cells, we think this is a good suggestion and will conduct the follow-up study.

2) Comment: Fig. S5a. The authors plotted the composition of the top 112 AS events in WT cells and a different set of top 112 AS events in the SF3B1R625H cells. This comparison does

not make sense. The authors should delete the WT column.

Author Response: Thank you for this thoughtful question. In Fig. S6a (Fig. S5a in the previous version), there was a significant difference in the types of alternative splicing events strongly associated with SF3B1 mutation compared with highly variable splicing events among samples without SF3B1 mutation. Thus, we believe the WT group is a necessary reference group for comparison, and similar analysis has been conducted in previous studies in the literature¹.

3) Comment: The binding of U2AF should be focused on the vicinity of the alternative exons instead of the entire gene body of the differentially spliced genes. No evidence in Fig. S7 supports the claim that “This suggested the impact of SF3B1R625H on the splicing of ESRRG through association with the U2 snRNP.”

Author Response: Thank you for this comment. We agree that the binding of the U2AF2 should work near the exons because the U2AF2 should be associated with the pre-mRNA on the polypyrimidine track at the end of the introns. Indeed, in Fig. S8b (Fig. S7b in the previous version), we scanned the U2AF2 binding motif at the ending 100bp of the annotated introns (i.e. 100bp upstream of the exons) from the reference transcriptome. The results showed that over half of the annotated introns were ending with the U2AF2 binding motif, which indicated that the splicing of known transcripts would be more likely impacted by the U2 snRNP complex. On the other hand, when splicing, the U2AF2 will recognize the binding motif on the whole pre-mRNA and then perform splicing. Hence, the more motifs enriched on the entire gene body, the more probable it is that abnormal unknown splicing events will

happen. As a conclusion, according to the motif analysis, not only will the known transcripts be affected by the U2 snRNP complex, but there will also be more uncertain splicing events occurring due to a large number of motifs enriched on the entire gene body.

We are sorry for the unclear statements. To emphasize these two points, we added some words in the main content and the legend of Fig S8.

Change to Text:

Line 161-163: *ESRRG* contained the most binding motifs (3088, significantly more than others), and over half of the annotated introns were ending with the U2AF2 binding motif (Supplementary Figure 8a, b).

Supplementary material: line 249-252: **b**, The count of introns containing the U2AF2 binding motifs on the last 100bp at the 3' ends in each transcript of *ESRRG*. Over half of the annotated introns are ending with the U2AF2 binding motif, which indicates that the known splicing of *ESRRG* would be more likely impacted by the U2 snRNP complex.

4) Comment: Fig. 3b. Cross-linking and Immunoprecipitation (CLIP) is a golden standard to determine the binding of an RNA binding protein on RNA. RIP signals can be caused by indirect interactions between proteins and RNA. Regardless, no SF3B1 mutant was included in this RIP experiment in Fig. 3b, and no description on the location of the primers for PCR. The PCR primers should be located at the vicinity of the branch point of the A3SS sites. It remains unclear whether the SF3B1 mutant has a higher binding affinity at the vicinity of the A3SS site of the *ESRRG*.

Author Response: Regarding CLIP vs RIP, the reviewer comment brings up the point of

possible non-specific results of RIP and suggests that more specific results can be obtained with CLIP. We thank the reviewer for this comment and agree that CLIP is an interesting experiment to perform for this interaction. However, RIP does confirm an interaction between the two targets which helps elucidate the understanding of our overall findings in this paper. RIP experiments have been used in numerous previous studies in the literature to demonstrate the interaction between splicing factors and RNA^{2,3,4}. In our study, the subsequent minigene reporter experiment led directly to the expected ESRRG abnormal splicing after infection of adenovirus expressing SF3B1^{R625H} (Supplementary Figure 9e), confirming that the R625H mutation of SF3B1 does induce aberrant splicing of ESRRG. In this study, our main finding is that the SF3B1 mutation status leads to aberrant splicing of ESRRG, which has downstream effect on PRL secretion in this subset of prolactinomas. While we think the CLIP experiment is interesting, it does not necessarily contribute further to our understanding of this study. We think that this experiment would take a significant amount of time and only add information regarding the specific location of the binding. We will certainly undertake CLIP to further define this interaction in a future study. Further, the reviewer asks for the primer locations. We can provide this information. Additionally, the reviewer comments that no SF3B1 mutant was used in the RIP experiment in figure 3b. That is correct. It was performed in figure 4b.

Change to Text:

Line 480-483: Primer pairs for quantitative real-time PCR were as follows: Canonical ESRRG: 5'-cgtggaggtcggcagaagtaca-3' (forward) and 5'-gcccatccaatgataaccacc-3' (reverse); Cryptic ESRRG: 5'-tgacagagtacgtggaggtcgg-3' (forward) and 5'-ctgcaggatcagaccagagcaat-3' (reverse).

The primer location please see below (highlighted in yellow):

Canonical ESRRG (1308 nt):

atgtcaacaaagatcgacacattgattccagctgttcctctcatcaagacggaacctccagcccagcctccctgacggacagcgtc
aaccaccacagccctgggtgctcttcagacgccagtgaggagctacagttcaacctgaatggccatcagaacggacttgactcgccac
ctctctacccttctgctcctatcctgggaggtagtgggctgtcaggaaactgtatgatgactgctccagcaccattgttgaagatcccca
gaccaagtgtgaatacatgctcaactcgatgcccaagagactgtgttagtgtgtggtgacatcgcttctgggtaccactatgggtagc
atcatgtgaagcctgcaaggcattctcaaggacaattcaaggcaatataagaatacagctgccctgccacgaatgaatgtgaatca
caaagcgcagacgtaaatcctgccaggctgccgctcatgaagtgtttaaagtgggcatgctgaagaaggggtgctcttgacag
agtagtggaggctggcagaagtacaagcgcaggatagatcgggagaacagcccatacctgaaccctcagctggttcagccagcca
aaaagccatataacaagattgtctcacattgttggtggctgaaccggagaagatctatgccatgcctgacccactgtccccgacagtg
acatcaaagccctcactacactgtgtgacttgccgaccgagagtggtggttatcattggatgggcgaagcatattccaggcttctcca
cgctgtccctggcggaccagatgagccttctgcagagtgttgatggaaatgatccttgggtcgtataccggtctcttctgtttgag
gatgaactgtctatgcagacgattatataatggacgaagaccagtccaaattagcaggccttctgatctaaataatgctatcctgcagct
ggtaaagaatacaagagcatgaagctggaaaaagaagaattgtcacctcaaagctatagctcttgctaattcagactccatgcacat
agaagatgtgaagccgttcagaagcttcaggatgtcttacatgaagcgtgcaggattatgaagctggccagcacatggaagaccctc
gtcgagctggcaagatgctgatgacactgccactcctgaggcagacctctaccaaggccgtgcagcatttctacaacatcaaactaga
aggcaaagtcccaatgcacaaactttttggaaatgttgaggccaaggtctga

5) **Comment:** The authors show that the expression levels of ESRRG are significantly higher in the SF3B1R625H cells than that in the WT cells. This suggests that the SF3B1R625H mutant promotes ESRRG transcription in addition to A3'SS. The description of the results was confusion.

Author Response: Thank you for this thoughtful question. The reviewer comments on the expression levels of ESRRG in Fig. 3c and draws the conclusion that the elevation in wild-type pituitary samples suggests that the SF3B1 mutant promotes ESRRG transcription in addition to alternative splicing. However, this experiment is not quantitative; it is qualitative in nature. Therefore, yes, the WT SF3B1 does result in some production ESRRG, as this is a native function of SF3B1. As shown in Fig.3g, when primary cultured pituitary tumor cells were infected with adenovirus expressing SF3B1^{WT} and SF3B1^{R625H} (lane 3 and lane 4) respectively, only the cryptic ESRRG increased significantly, while the canonical ESRRG did not significantly increase.

6) Comment: Fig. 4b. The authors showed in Fig. 3b that there was a 3-fold increase in WT SF3B1 binding to ESRRG. However, this enrichment was not observed in Fig. 4b (see left two bars, also note the label, it should be IgG and SF3B1). How to explain this discrepancy? Note that the reviewer's comment is based on current figures. PCR primers need to be located in the vicinity of the A3'SS. Together, the conclusion that "This confirmed that the mutant SF3B1 would trigger the aberrant splicing in ESRRG." Is not valid.

Author Response: Thank you for this opportunity to clarify these figures. In Fig. 3b, we detected the interaction between SF3B1 and canonical ESRRG mRNA in common pituitary tumors. In Fig. 4b, we detected the interaction between SF3B1 and cryptic ESRRG mRNA in SF3B1 wild-type and mutant pituitary tumors. Canonical ESRRG does not exist in SF3B1 wild-type pituitary tumors, so there is no comparability with Fig. 3b. Further, the reviewer asks for the primer locations. We can provide this information below.

Change to Text:

Line 480-483: Primer pairs for quantitative real-time PCR were as follows: Canonical

ESRRG: 5'-cgtggaggtcggcagaagtaca-3' (forward) and 5'-gcccatccaatgataaccacc-3' (reverse);

Cryptic ESRRG: 5'-tgacagagtacgtggaggtcgg-3' (forward) and 5'-ctgcaggatcagaccagagcaat-3' (reverse).

The primer location please see below (highlighted in yellow, the extra 21bp in Cryptic ESRRG is shown in red):

Cryptic ESRRG (1413 nt):

atgtggcgagaatgtgattggggtcttgagcagtcagtctgatctggcctgtgttcctcagctaaaaggcttctgcagaatgtcaa
acaaagatcgacacattgattccagctgttcgtccttcatcaagacggaacctccagcccagcctcctgacggacagcgtcaaccac
cacagccctgggtgctcttcagacgccagtgggagctacagttcaacctgaatggccatcagaacggacttgactcgccacctctta
cccttctgctcctatcctgggaggtagtgggcctgtcaggaaactgtatgatgactgctccagcaccattgtgaagatccccagacaa
gtgtgaatacatgctcaactcgatgccaagagactgttttagtgtgtgtgacatcgcttctgggtaccactatggggtagcatcatgt
gaagcctgcaaggcattctcaagaggacaattcaaggcaatatagaatacagctgccctgccacgaatgaatgtgaaatcaciaagc
gcagacgtaaatcctgccaggctgccctcatgaagtgtttaaagtggcatgctgaaagaaggggtgcgtcttgacagagtacgt
ggaggtcggcagaagtacaagcgcaggatagatcgggagaacagccatacctgaaccctcagctggtcagccagccaaaaagc
cattgctctggctgatcctgcagataacaagattgtctcacattgttggtggctgaaccggagaagatctatgccatgcctgacctact
gtccccgacagtgacatcaaagccctcactacactgtgtgacttgccgaccgagagttggtggttatcattggatgggccaagcatatt
ccaggcttccacgctgtcctcggggaccagatgagccttctgcagagtgttgatggaatttgatccttgggtcgtataccggt
ctcttctgttgaggatgaactgtctatgcagacgattatataatggacgaagaccagtccaattagcaggccttctgatctaaataatg
ctatcctgcagctggtaaagaaatacaagagcatgaagctggaaaaagaagaattgtcacccctcaaagctatagctcttgctaattcag
actccatgcacatagaagatgtgaagccgtcagaagcttcaggatgtttacatgaagcgtgcaggattatgaagctggccagcac

atggaagaccctcgtcgaactggcaagatgctgatgacctgccactcctgaggcagacctctaccaaggccgtgcagcatttctaca
acatcaaactagaaggcaaagtccaatgcacaaactttttggaaatggtggaggccaaggtctga

7) Comment: Fig. S10. Western blot of WT and cryptic ESRRG expression needs to be shown to make sure that the expression levels of WT ESRRG and the cryptic ESRRG are comparable.

Author Response: Thanks for this comment. The corresponding western blot has been supplemented in Fig. S11 (Fig. S10 in the previous version) according to the requirements of the reviewer.

8) Comment: Fig. 4d. Results showed that KD of ESRRG caused a roughly 2-fold decrease in PRL level. Should it be expected that ectopically expressing WT ESRRG should rescue the decreased PRL expression, which was not the case seen in Fig. S10.

Author Response: Thanks for this comment. First of all, please allow me to apologize for our mistakes in Fig. S11 (Fig. S10 in the previous version), both canonical and cryptic ESRRG resulted in a significant augment of PRL secretion, but in the canonical ESRRG group, we forgot to mark "###" in the figure, but the corresponding figure legend is correct. We have corrected it.

9) Comment: SF3B1R625H was previously identified as a prevalent SF3B1 mutant in a few published papers. The authors should acknowledge previous findings when they first describe SF3B1R625H in line 109-111. Examples are:

- Recurrent mutations at codon 625 of the splicing factor SF3B1 in uveal melanoma. *Nat Genet.* 2013 Feb; 45(2): 133–135.

- Cancer-associated SF3B1 mutations affect alternative splicing by promoting alternative branchpoint usage. *Nature Communications* volume 7, Article number: 10615 (2016)

- Recurrent hotspot SF3B1 mutations at codon 625 in vulvovaginal mucosal melanoma identified in a study of 27 Australian mucosal melanomas. *Oncotarget.* 2019; 10:930-941.

Author Response: We will provide additional references according to the requirements of the reviewer.

Change to Text:

Line 110-111: Mutations in SF3B1 R625 have been described in uveal melanoma, vulvovaginal mucosal melanoma, and other cancers^{5, 6, 7}.

1. Wang L, *et al.* Transcriptomic Characterization of SF3B1 Mutation Reveals Its Pleiotropic Effects in Chronic Lymphocytic Leukemia. *Cancer Cell* **30**, 750-763 (2016).
2. Wan L, *et al.* SRSF6-regulated alternative splicing that promotes tumour progression offers a therapy target for colorectal cancer. *Gut* **68**, 118-129 (2019).
3. Tarallo R, *et al.* The nuclear receptor ERbeta engages AGO2 in regulation of gene transcription, RNA splicing and RISC loading. *Genome Biol* **18**, 189 (2017).
4. Xin R, *et al.* SPF45-related splicing factor for phytochrome signaling promotes photomorphogenesis by regulating pre-mRNA splicing in Arabidopsis. *Proc Natl Acad Sci U S A* **114**, E7018-E7027 (2017).
5. Harbour JW, Roberson ED, Anbunathan H, Onken MD, Worley LA, Bowcock AM. Recurrent mutations at codon 625 of the splicing factor SF3B1 in uveal melanoma. *Nat Genet* **45**, 133-135 (2013).

6. Alsafadi S, *et al.* Cancer-associated SF3B1 mutations affect alternative splicing by promoting alternative branchpoint usage. *Nat Commun* **7**, 10615 (2016).
7. Quek C, *et al.* Recurrent hotspot SF3B1 mutations at codon 625 in vulvovaginal mucosal melanoma identified in a study of 27 Australian mucosal melanomas. *Oncotarget* **10**, 930-941 (2019).

REVIEWERS' COMMENTS:

Reviewer #4 (Remarks to the Author):

The authors have address all of the significant comments of the previous review with significant additional data that greatly strengthen the manuscript. The Western blots of SF3B1 that are in supplement (i.e. for Fig 2) should be moved to the main manuscript as these are critical to the interpretation of the data.

Reviewers' comments:

Reviewer #4:

Comment: The authors have address all of the significant comments of the previous review with significant additional data that greatly strengthen the manuscript. The Western blots of SF3B1 that are in supplement (i.e. for Fig 2) should be moved to the main manuscript as these are critical to the interpretation of the data.

Author Response: We appreciate your suggestion. We have updated the corresponding Fig 2.